# LLM AGENTS FOR LITERATURE TO CODE CONVERSION: CASE STUDY OF HEAT EXCHANGER DESIGN

**Sandeep Mishra**
Tata Research Development and Design Centre
Pune, Maharashtra, India
`sandeepm.mishra@tcs.com`

**Vishal Jadhav**
Tata Research Development and Design Centre
Pune, Maharashtra, India
`vi.suja@tcs.com`

**Shirish Karande**
Tata Research Development and Design Centre
Pune, Maharashtra, India
`shirish.karande@tcs.com`

**Venkataramana Runkana**
Tata Research Development and Design Centre
Pune, Maharashtra, India
`venkat.runkana@tcs.com`

## ABSTRACT

This paper introduces a framework that utilizes large language model (LLM) agents to extract and convert mathematical models from engineering literature into executable code. Autonomous or semi-autonomous conversion of literature into code facilitates downstream tasks such as hypothesis generation, verification, and benchmarking. Focusing on heat exchanger design, our approach efficiently integrates model extraction, code generation, and performance optimization, with minimal human intervention. The system's knowledge base is continuously refined with each new paper, leading to ongoing improvements. Experiments conducted on 115 research articles using the HxAgent approach demonstrate substantial improvements over the previous non-agentic baseline, HxLLM. Although the work is still in progress, the results highlight the potential of agent-driven workflows in advancing scientific discovery.

## 1 INTRODUCTION

Heat exchangers are a crucial component in various industries, including chemical processing, power generation, HVAC systems, automotive, and renewable energy, where efficient heat transfer is essential for improving energy efficiency, reducing operational costs, and promoting environmental sustainability. However, the design and optimization of heat exchangers pose significant challenges, particularly in complex and demanding environments such as power plants, where issues like fouling can significantly reduce heat transfer efficiency.

The design and optimization of heat exchangers as shown in the Figure 5, involve a series of systematic steps, including problem definition, selection of a mathematical model, optimization, validation, testing, refinement, and final implementation. The core of the process lies in the selection of mathematical models that describe heat transfer mechanisms and the identification of optimal design parameters by evaluating various heat transfer metrics in conjunction with corresponding design constraints. The literature on heat exchanger design and optimization encompasses various methods aimed at improving performance, cost-effectiveness, and energy efficiency (see details on Appendix section B). In addition to the basic model, the optimization process involves different algorithms such as PSO, Genetic Algorithms (GA), and other techniques (see Appendix section - C) to further enhance the cost function optimization.

The use of Large Language Models (LLMs) and AI agents in design and optimization is changing engineering practices, creating new possibilities for improving the performance and efficiency of industrial equipments (see Appendix section - A). Inspired by these advancements, we propose an LLM based multi-agent system, 'HxAgent', for optimizing industrial heat exchangers. This approach is inspired by the principles of human collaboration and decision-making, where agents autonomously make decisions, adapt to dynamic conditions, and collaborate to optimize heat exchanger designs more effectively.

The evaluation of Large Language Model (LLM)-generated code has been explored in several recent studies see details in Appendix section - A.1. These studies collectively motivate the development of our evaluation framework. We also compare the performance of our proposed HxAgent framework with the with HxLLM framework (Mishra et al., 2024) that provided the considered dataset. HxAgent consistently improves the overall quality of the generated code. Our research has significant implications for various industries, including chemical processing, power generation, and renewable energy, where efficient heat transfer is critical for improving energy efficiency and reducing operational costs.

## 2 HXAGENT FRAMEWORK

The HxAgent framework introduces an approach that integrates self-reflection, human-in-the-loop (HITL) interaction, and RAG techniques. Designed to enhance decision-making processes, this framework iteratively refines outcomes by incorporating expert feedback, optimizing results, and continuously reassessing them using evaluation metrics. At its core, the framework utilizes a Self-Reflection Information Dataset (see Appendix - F), which includes a collection of mathematical models, optimization algorithms, and related processes, providing a robust foundation for ongoing improvement and adaptation. To perform the design and optimization using HxAgent framework, we have divided the process in subprocess like

1. Mathematical model identification
2. Code generation for mathematical model and optimization algorithm
3. Code refinement and validation

Each of these subprocess tasks are performed by set of agents ( described in Appendix E) and repositories (described in Appendix F) are explained below:

**Mathematical Model Identification:** The HxAgent framework uses a summary creator agent $(A_S)$ to extract mathematical models from research articles. The TF-IDF Agent $(A_T)$ in the HxAgent framework enhances this process by comparing the extracted model with existing models in the repository, improving the accuracy of model matching and retrieval of relevant code.

**Code Generation and Optimization:** In the HxAgent framework, the Planner Agent $(A_P)$ and Designer Agent $(A_D)$ collaborate to plan and build the mathematical code, while the Optimizer Agent $(A_O)$ independently generates optimization algorithm. In the HxLLM framework, optimization code was created alongside the mathematical model. However, in this case, there is a dedicated agent that generates the optimization code independently, making it easier to analyze the code separately in later stages.

**Self-Reflection and Code Refining:** The Code Refiner Agent $(A_C)$ in the HxAgent Framework introduces a self-reflection mechanism that reviews and suggests improvements for generated code based on a self-reflection dataset. This mechanism enhances the quality of code refinement and reduces the need for human intervention, a feature not present in HxLLM frameworks.

**Code Correction and Validation:** The code generated by the HxAgent is executed, and if the execution results in an error, the RAG-based technique is used to find a similar error and its resolution. The Code Correction Agent $(A_{CC})$ and Validator Agent $(A_V)$ in the HxAgent framework add an extra layer of quality control to ensure the final optimized code is accurate and functional. This systematic approach includes dedicated agents for validation, providing a thorough and reliable process for code correction.

The pseudo-code of the whole HxAgent framework is illustrated in algorithm 1, which outlines a comprehensive process for optimized code generation, involving multiple agents and steps to ensure the creation of accurate and efficient code (illustrated in Figure 1). For this study, we have considered research article $(P)$ as source of mathematical model and optimization algorithm, however in practice, this can also be obtained from user. The process begins with the Summary Creator Agent $(A_S)$, which generates a summary $(S)$ of the mathematical models in a given research article $(P)$ using a large language model. The TF-IDF Agent $(A_T)$ then compares this summary $(S)$ with existing mathematical model summaries $(S_1, S_2, S_3, \ldots)$ stored in the mathematical model repository $M$, where each $S_i$ represents the summary of a mathematical model from articles in the repository. It calculates a similarity score, and if the score exceeds 0.75, the Planner Agent-1 $(A_{P1})$ and Designer

Agent-1 ($A_{D1}$) are engaged to generate the mathematical model code. Otherwise, the Planner Agent-2 ($A_{P2}$) and Designer Agent-2 ($A_{D2}$) take part in an alternative planning process, which also results in the creation of mathematical model code. The Optimization Agent ($A_O$) generates optimization algorithm code independently, which is then refined and merged with the mathematical model code by the Code Refiner Agent ($A_C$) with human in the loop(HITL) and Code Merger Agent ($A_M$), respectively. Finally, it checks if the execution of the output finalised code ($FC$) results in an error ($Er$). If the error is identified as "RAG-E" (Errors that already stored in the RAG) , then the error is considered complex and has already been solved and stored in the code error repository. In this case, the code correction agent ($A_{CC}$) is invoked, using the error ($Er$) and its corresponding solution ($RAG - S$) (solution to those RAG-E errors that already stored in the RAG) as a reference. Otherwise, the code is corrected using only the error ($Er$). If the corrected code executes successfully, the output code ($OC$) is accepted as correct, and the process ends without needing further correction. This iterative process ensures the generation of high-quality, optimized code that meets the required standards.

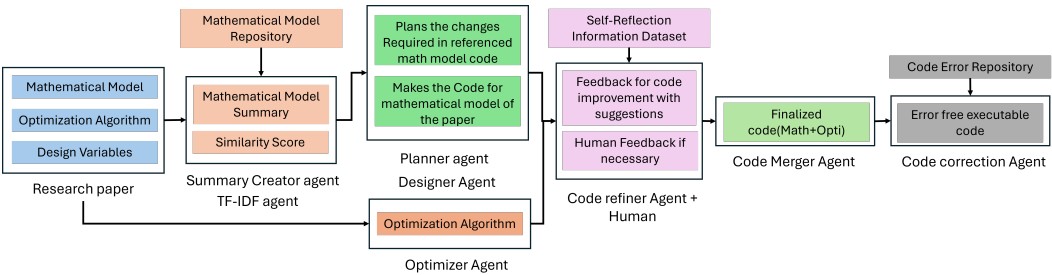

Figure 1: Algorithm of Agentic Framework

## 2.1 EVALUATION METHOD

A multifaceted evaluation framework was developed to assess the quality of LLM and agent-generated code based on six criteria: Accuracy, Functionality, Completeness, Readability, Robustness, and Maintainability. The code was scored on a scale of 1 to 5 across these criteria, evaluated by multiple reviewers on 115 articles. Specific evaluation metrics were defined for each criterion, and further details are available in the Appendix section - D.

## 3 RESULTS

The HxAgent framework processes user inputs (research articles) through four key steps: model identification, optimization algorithm generation, self-reflection, and final output generation with code correction. This framework was tested on 115 research articles, utilizing the Llama 3.1 model as the language model and the Lang-graph (lan) framework as the agentic structure. In order to compare performance with a non-agentic baseline, the HxLLM framework was re-implemented on the same 115 articles by using Llama 3.1 model.

The evaluation of 115 research articles across the two frameworks, HxLLM and HxAgent, revealed varying performance levels on six criteria: accuracy, functionality, completeness, readability, robustness, maintainability, and overall evaluation. Group 1 and Group 3 emerge as the top performers (illustrated in Figure - 2) due to their comprehensive data and well-structured methodologies. These groups are marked by nearly complete data sets, and where there is missing data, we have developed optimization algorithms, variables, and mathematical codes to compensate for the gaps. This enables a more complete analysis, which contributes significantly to their high ratings in most of the evaluation criteria.

In this section we describe our observations. Additionally, to enable subjective judgment, Appendix section G demonstrates how the HxAgent framework outperforms the HxLLM framework by providing examples from three different papers. It illustrates how HxAgent improves the correct selection of variables, ensures the correct selection of optimization algorithms, and enhances functionality

---

**Algorithm 1** Optimized Code Generation

---

**while** true **do**

    Input: Research article $P$

    **Summary Creation**: $A_S(P) \to S$

    **TF-IDF-based Retrieval**: $A_T(S, M) \to T$ (Compares $S$ with $S'$ present in $M$ and gives a score $T$)

    **Mathematical Model Code Generation**:

    **if** $T > 0.75$ **then**

        **Path-1: Similar model found**

        $P_1 = A_{P1}(P)$

        $D_1 = A_{D1}(P_1)$

    **else**

        **Path-2: No similar model found**

        $P_2 = A_{P2}(P)$

        $D_2 = A_{D2}(P_2)$

    **end if**

    **Optimization algorithm code generation**: $A_O(P) \to O$

    **Code Refining with HITL**: $A_C(D, O) \to C$ (Where $D$ is either $D_1$ or $D_2$)

    **Code Merging**: $A_M(D, O, C) \to FC$ (Final Optimized Code)

    **Code Correction:** $Er = \text{Exec}(FC)$

    **if** $Er = \text{RAG-E}$ **then**

        $OC = A_{CC}(Er, RAG - S)$ {(Where RAG-S is the solution corresponding to error RAG-E)}

    **else**

        $OC = A_{CC}(Er)$

    **end if**

    **if** $\text{Exec}(OC) = \text{Code output}$ **then**

        $OC = FC$ {Code is already correct, no need to enter the code correction loop}

    **end if**

    **Validation**: $A_V(OC) \to V$

**end while**

---

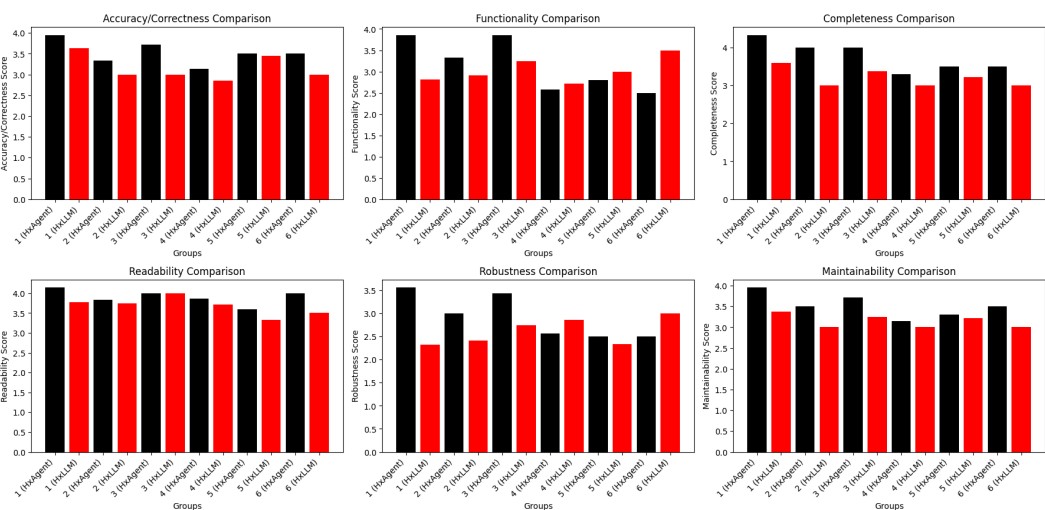

Figure 2: Comparison of HxAgent and HxLLM Frameworks Across Multiple Metrics

and robustness. Meanwhile, in Appendix H, we provide an example from one paper to illustrate the step-by-step process of how the HxAgent framework operates.

It was evident that the HxAgent framework consistently outperforms the HxLLM framework across most metrics, especially in accuracy, functionality, and maintainability. For instance (refer Figure

- 2), Group 1 (Agent: 3.953 vs. HxLLM: 3.628), Group 3 (Agent: 3.714 vs. HxLLM: 3.000), and Group 5 (Agent: 3.500 vs. HxLLM: 3.444) exhibit clear advantages in these areas. The Agent framework also shows higher ratings in Completeness and Readability, demonstrating that it offers a more comprehensive and understandable approach to solving complex problems.

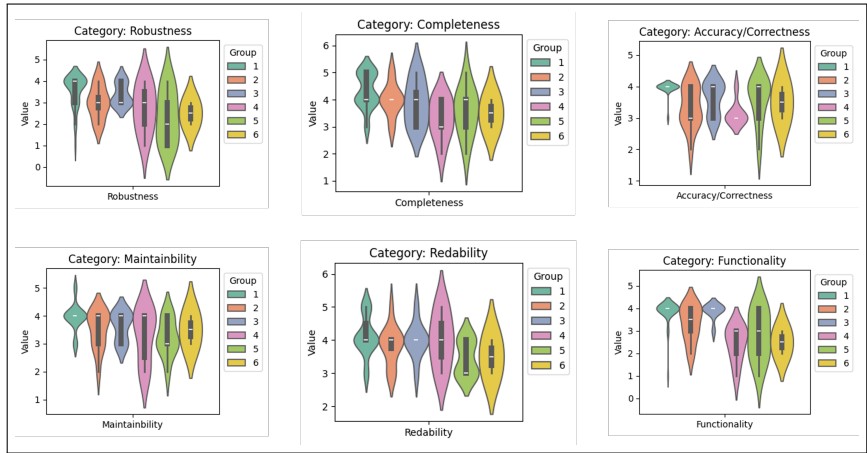

Figure 3: Score of HxAgent Framework Across All Groups and Parameters

In contrast, the HxLLM framework shows weaker performance, particularly in accuracy and functionality, with noticeable gaps in Groups 1 and 3. The HxAgent framework's superior handling of data integration, optimization algorithms, and mathematical models leads to more accurate and functional outcomes. This is likely due to the HxAgent's better data structuring and model development capabilities, while the HxLLM framework appears limited in data handling and adaptability. From the Figure 3 it can be seen that Groups 2, 4, 5, and 6 face distinct challenges. Group 4, for instance, struggles with the complex structure of heat exchanger networks, and many articles fail to define these structures adequately, resulting in lower evaluation scores. Group 2's reliance on simulation tools such as CFD complicates the evaluation process. In Groups 5 and 6, models are often insufficiently validated or rely on incomplete datasets, which negatively impacts their performance in accuracy and robustness. The key points can be summarized as follows: **Inconsistent Methodologies**, where many articles in these groups use non-standard or hard-to-replicate methods, leading to variability and reduced reliability; and **Limited Practical Application**, as the lack of real-world data integration and insufficient testing under practical conditions prevent these groups from achieving the same levels of maintainability and functionality as Groups 1 and 3. Further details on limitations and future work can be found in Appendix J.

In conclusion, Groups 1 and 3 outperform others due to well-structured methodologies and nearly complete datasets, while Groups 2, 4, 5, and 6 are hindered by complex requirements, undefined structures, and inconsistent methodologies.

## 4 CONCLUSION

This study introduced the HxLLM HxAgent frameworks, which utilize nine specialized agents powered by Large Language Models (LLMs) to automate the design and optimization of heat exchangers. By integrating key mechanisms such as mathematical model extraction, code generation, and error correction via Retrieval-Augmented Generation (RAG), self-reflection, and human-in-the-loop processes, our approach demonstrated the ability to produce error-free code for 115 research articles across six different groups. The HxAgent framework achieved an impressive overall score based on a comprehensive six-metric evaluation, outperforming the HxLLM framework. Our findings highlight that the HxAgent framework, with its advanced automation capabilities, holds significant potential for automating the complex design and optimization tasks of heat exchangers. This could lead to substantial improvements in energy efficiency and cost reduction in the process industry. However, the study also revealed limitations within the current framework, particularly in areas involving data completeness and the need for external simulations, where the framework often falls

short. Looking ahead, future research should focus on enhancing the agent's ability to parse and process complex information from various document formats. Additionally, reducing the dependency on user input and improving integration with external data repositories and simulation tools will further optimize the framework's capabilities, enabling more robust and comprehensive automation in heat exchanger design and optimization.

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

# A   LITERATURE REVIEW ON APPLICATION OF LLM AND AI AGENTS ON DESIGN AND APPLICATIONS OF DIFFERENT ENTITIES

For example Ni et al.(Ni & Buehler, 2023) introduces MechAgents, a platform where LLM agents collaborate autonomously to solve mechanics problems using finite element methods. This system, designed for solving elasticity problems, integrates the intelligence of language models with the reliability of physics-based modeling.Guo et al. (Guo et al., 2024) proposes ControlAgent, a framework that integrates LLM agents with control system domain expertise to automate the controller design process. This system is capable of gradually refining controller parameters to meet specific requirements for stability, performance, and robustness. Jadhav et al. (Jadhav & Farimani, 2024) presents an approach where pre-trained LLMs are integrated with a Finite Element Method (FEM) module for mechanical design (truss structure) optimization. This system allows LLMs to autonomously generate and refine designs from natural language specifications demonstrating the potential of LLMs to streamline the design process. Tao et al.'s (Tao et al., 2024) MAGIS framework focuses on using LLM agents to resolve issues in software repositories. The system leverages multiple types of agents to enhance LLM performance, achieving significant improvements in issue resolution rates. Liu et al. (Liu et al., 2024a)introduces LayoutCopilot, a framework that empowers LLMs to assist in analog layout design by converting natural language instructions into executable script commands. This system simplifies the interaction between designers and tools. Ma et al. (Ma et al., 2023) explores the use of LLMs in conceptual design, emphasizing how these models generate feasible and useful solutions. The study compares LLM-generated solutions to crowdsourced ones, showing that LLMs can produce designs that are more feasible and useful, while crowdsourced solutions tend to be more novel.Zheng et al. (Zheng et al., 2023) highlights how LLMs have shown significant potential in transforming scientific discovery, offering solutions to complex problems like molecular property prediction and scientific data interpretation.

Gao and Li (Liu et al., 2024b) introduce AgentHPO, a system leveraging LLMs for autonomous hyperparameter optimization. By iterating on historical trial data, this approach reduces the number of experiments required, simplifies the setup process, and improves interpretability. Empirical results show that AgentHPO often outperforms human-driven optimization.Mostajabdaveh (Mos) presents a framework for converting natural language descriptions into optimization models. This framework uses multiple LLM agents to autonomously generate and verify models, surpassing traditional methods in efficiency and accuracy. Pluhacek et al. (Pluhacek et al., 2023) explores the use of LLMs, particularly GPT-4, to create hybrid swarm intelligence algorithms for optimization tasks. The approach showcases the potential for LLMs to generate innovative and effective optimization algorithms. Ma et al,(Ma et al., 2024) introduces LLaMoCo, a framework for tuning LLMs for optimization tasks. This method reduces sensitivity to prompt design, improving performance across various optimization challenges. OptiMUS is introduced by Ahmadi Teshnizi et al.(AhmadiTeshnizi et al., 2023), using LLMs to solve linear programming problems. The system iteratively generates and refines solutions, outperforming human-designed prompts. Chen et al. (Chen et al., 2023) present OptiChat, which is a system designed to diagnose infeasible optimization models. It identifies sources of infeasibility and offers suggestions to make models feasible, helping optimize the process. Finally, Erikstad (Erikstad, 2024) explores the integration of multi-agent LLMs with Model-Based Systems Engineering (MBSE) to streamline the development of design optimization models, significantly improving the efficiency of the optimization process. The reviewed studies demonstrate that LLMs and AI agents can significantly enhance optimization and design tasks. They can automate the creation of optimization models, improve convergence during optimization, generate novel solutions, and diagnose design issues. HxLLM framework proposed by Mishra et al.(Mishra et al., 2024) leverages LLMs to generate mathematical models and optimization algorithms based on user inputs. However, this framework has limitations that hinder its overall performance. Firstly, it requires human intervention at various stages, reducing its autonomy. The code generation process depends on the quality of the input prompts, which can lead to inconsistencies in the output. Additionally, the framework lacks access to external datasets, limiting its ability to generate solutions beyond the provided articles. It also lacks a self-reflection mechanism to evaluate and improve its output, increasing the likelihood of errors. The framework struggles with selecting appropriate optimization algorithms, addressing functionality issues, and making accurate variable selections. Furthermore, the generated results are not evaluated for accuracy or effectiveness, which compromises the reliability and robustness of the solutions. These limitations highlight key areas for improvement in the framework's development.

### A.1 Literature on Code Evaluation

Cano et al. (Çano & Bojar, 2020) introduced an automated human-likeliness score for natural language generation, highlighting the potential of LLMs to automate complex evaluations. Chen et al. (Chen et al., 2024) emphasized the need for a multi-dimensional evaluation of LLM performance, suggesting that combining metrics like code correctness, efficiency, and readability provides a more comprehensive assessment. Tong et al. (Tong & Zhang, 2024) presented CODEJUDGE, a framework that uses LLMs to evaluate semantic correctness without test cases, emphasizing the importance of deeper, more reliable evaluations.

## B Heat Exchanger Design and Optimization Methods: Techniques, and Optimization Algorithms

This study categorizes the approaches into six main groups(see Figure 6):

1. Group - 1 : Minimization of Cost Based on Geometrical Parameters Using a Mathematical Model and Optimization of Heat Exchanger Design Using Nature-Inspired Techniques
2. Group - 2 :Thermal Design and Simulation
3. Group - 3 :Exergy Analysis and Life Cycle Assessment
4. Group - 4 :Optimization of Heat Exchanger Networks
5. Group - 5 :Hybrid Mathematical Models
6. Group - 6 :Minimization of Life Cycle Irreversibility and Maximization of Heat Transfer

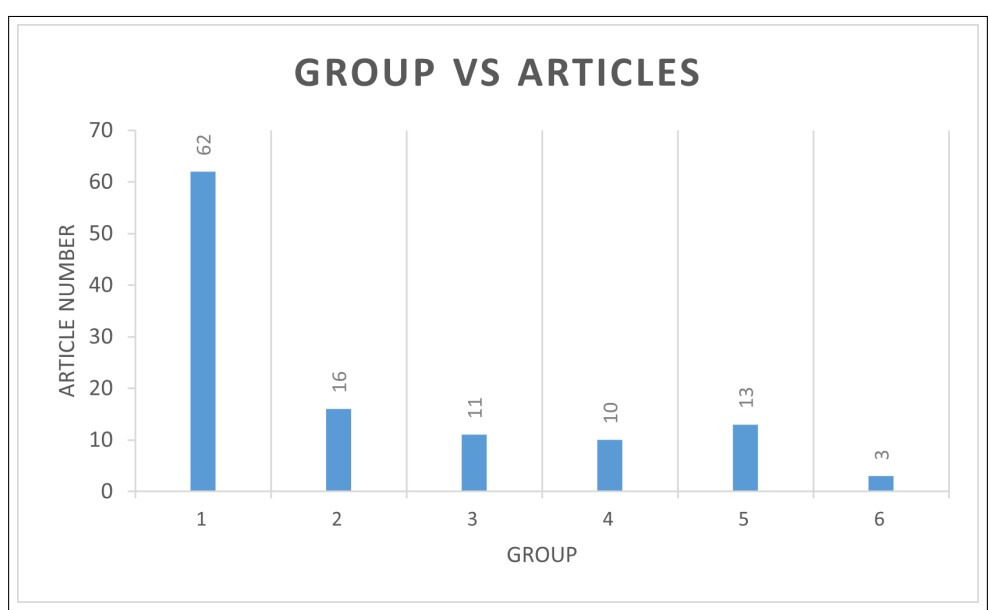

Figure 4: Groups vs Number of articles

Among these, the primary focus is Group 1 having the highest number of research articles (see Figure 4), which involves minimizing costs based on geometrical parameters using a mathematical model and optimizing heat exchanger design with nature-inspired techniques.

The mathematical model for heat exchanger design begins with the calculation of the tube-side heat transfer coefficient $(ht)$ in a shell and tube heat exchanger. This coefficient is computed using correlations based on the Reynolds number $(Re_t)$ and Prandtl number $(Pr_t)$, with different formulas for laminar, turbulent, and very turbulent flow regimes. The flow velocity $(v_t)$ is derived from the fluid properties and tube dimensions, while the number of tubes $(N_t)$ is calculated using an empirical relation based on the tube pitch and number of passes. For the shell side, the heat transfer

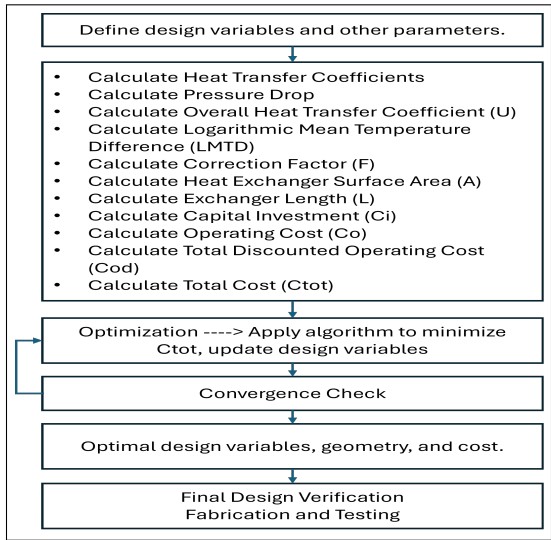

Figure 5: Design and Optimization steps of a heat exchanger

coefficient ($h_s$) is calculated using Kern's formulation, considering fluid properties and the hydraulic diameter. The overall heat transfer coefficient ($U$) is derived by combining both the shell and tube-side heat transfer coefficients, fouling resistances, and the tube diameter ratio. The logarithmic mean temperature difference (LMTD) is used to evaluate temperature variation across the heat exchanger, with a correction factor (F) applied based on the flow configuration. The heat exchanger surface area (A) is determined from the heat transfer rate and the overall heat transfer coefficient (U), while pressure drop calculations for both the tube and shell sides are performed to ensure an efficient system design. The optimization process focuses on minimizing the total cost, which includes capital investment, energy costs, and annual operating costs. Capital investment is directly related to the heat exchanger surface area, and the total discounted operating cost accounts for pumping power to overcome friction losses. In addition to the basic model, various heat transfer models and pressure drop correlations are considered, including Nusselt number-based and flow regime-based approaches for the tube side, as well as Kern's method and Bell-Delaware method for the shell side.

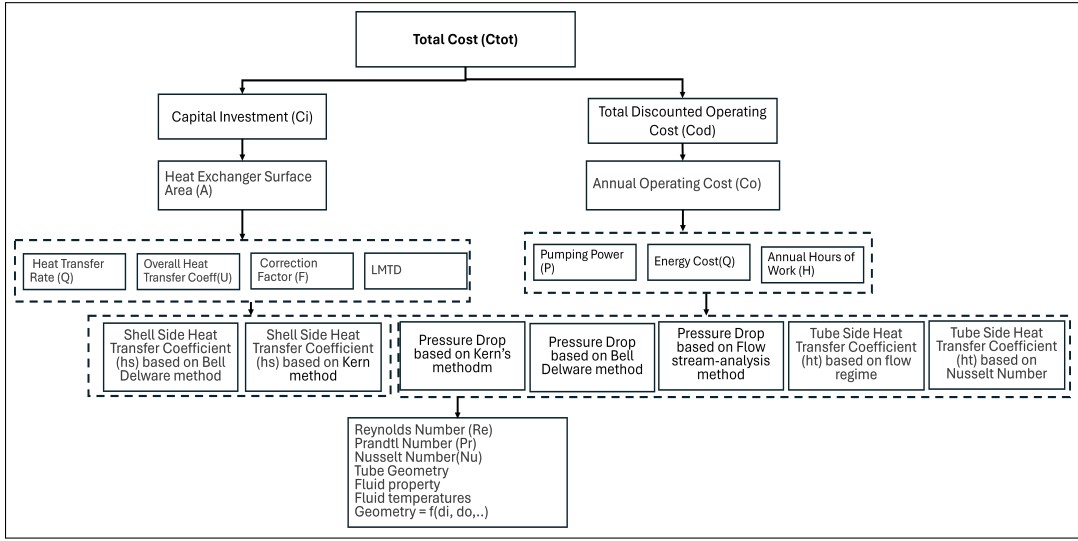

Figure 6: Design and Optimization steps of a heat exchanger of Group -1

## B.1 MODELLING EQUATIONS

The equations used for modeling are provided in Table 1.

Table 1: Literature on Heat Exchanger design and optimization - equations used for modelling

| Design Category | Design Variable | Types | Description | Illustrative PDE/Equation |
|---|---|---|---|---|
| Tube side Reynolds number | $Re_t$ | | | $$Re_t = \frac{\rho_t \cdot v_t \cdot d_i}{\mu_t}$$ Serth & Lestina (2014) |
| Prandtl's number of the tube | $Pr_t$ | | | $$Pr_t = \frac{\mu_t \cdot C_{p_t}}{k_t}$$ Serth & Lestina (2014) |
| number of tubes | $N_t$ | | $n_1$ is the number of tube passes,C and $n_1$ are coefficients that are taking values according to flow arrangement and number of passes | $$N_t = C \cdot \left(\frac{D_s}{d_o}\right)^{n_1}$$ Serth & Lestina (2014) |
| Tube Inside Diameter | $d_i$ | | | $$d_i = 0.8 \cdot d_o$$ Serth & Lestina (2014) |
| Flow velocity for tube side | $v_t$ | | | $$v_t = \frac{m_t}{\left(\frac{\pi}{4}\right) \cdot d_i^2 \cdot \rho_t \cdot \frac{N_t}{n}}$$ Serth & Lestina (2014) |
| Nusselt Number | $Nu$ | | | $$Nu = 0.023 \cdot Re^{0.8} \cdot Pr^{0.4}$$ Lahiri & Khalfe (2015),Selbaş et al. (2006) |
| Darcy Friction Factor | $f$ | | | $$f = (1.82 \cdot \log_{10}(Re) - 1.64)^{-2}$$ Serth & Lestina (2014) |
| Tube Side Heat Transfer Coefficient | $h_t$ | Based on Nusselt number | | $$h_{\text{tube}} = \frac{Nu \cdot k}{d_i}$$ Lahiri & Khalfe (2015),Selbaş et al. (2006),? |
| Tube side heat transfer coefficient | $h_t$ | Based on flow regime | For $Re_t < 2300$ (laminar flow) | $$h_t = \frac{k_t}{d_i}\left(3.657 + \frac{0.0677 \cdot \left(Re_t \cdot Pr_t \cdot \left(\frac{d_i}{L}\right)^{1.33}\right)^{0.33}}{1 + 0.1 \cdot Pr_t \cdot \left(Re_t \cdot \left(\frac{d_i}{L}\right)\right)^{0.3}}\right)$$ |
| | | | For $2300 < Re_t < 10000$ (transitional flow) | $$\text{Numerator} = \frac{f_t}{8} \cdot (Re_t - 1000) \cdot Pr_t \cdot \left(1 + \left(\frac{d_i}{L}\right)^{0.67}\right)$$ $$\text{Denominator} = 1 + 12.7 \cdot \left(\frac{f_t}{8}\right)^{0.5} \cdot \left(Pr_t^{0.667} - 1\right)$$ $$h_t = \frac{k_t}{d_i} \cdot \frac{\text{Numerator}}{\text{Denominator}}$$ |

| Design Category | Design Variable | Types | Description | Illustrative PDE/Equation |
|---|---|---|---|---|
| | | | For $Re_t > 10000$ (turbulent flow) | $$h_t = 0.027 \cdot \frac{k_t}{d_i} \cdot \left(Re_t^{0.8}\right) \cdot \left(Pr_t^{0.33}\right) \cdot \left(\frac{\mu_t}{\mu_{\text{wt}}}\right)^{0.14}$$ Serth & Lestina (2014) |
| Reynolds number for shell side | $Re_s$ | | | $$Re_s = \frac{m_s d_e}{\mu_s A_s}$$ Serth & Lestina (2014) |
| Prandtl number for shell side | $Pr_s$ | | | $$Pr_s = \frac{\mu_s C_{p_s}}{K_s}$$ Serth & Lestina (2014) |
| Flow velocity for the shell side | $v_s$ | | | $$\text{v}_s = \frac{m_s}{\rho_s A_s}$$ Serth & Lestina (2014) |
| Shell Side Heat Transfer Coefficient | $h_{\text{shell}}$ | Kern's Method | Kern's formulation for segmental baffle shell-and-tube exchanger | $$h_{\text{shell}} = \frac{k}{D_e} \cdot 0.36 \cdot Re^{0.55} \cdot Pr^{\frac{1}{3}} \cdot \left(\frac{\mu_{\text{shell}}}{\mu_{\text{tube}}}\right)^{0.14}$$ Serth & Lestina (2014) |
| | $h_{\text{shell}}$ | Bell-Delaware method | ideal cross-flow heat-transfer coefficient ($hc$) Baffle window flow CF ($Jc$) Baffle leakage CF ($JL$) Tube bundle bypass CF ($JB$) Unequal baffle spacing CF ($JS$) Laminar flow CF ($JR$) Wall viscosity ($J$) | $$h_s = h_c J_C J_L J_B J_S J_R J_\mu$$ $$h_c = j_i c_p G Pr^{-\frac{2}{3}}$$ $$j_i = a_1 \left(\frac{1.33}{\frac{L_{tp}}{D_t}}\right)^a Re^{a_2}$$ $$a = \frac{a_3}{1 + 0.14 Re^{a_4}}$$ |
| Equivalent Diameter for Square Pitch | $D_{\text{eq}}$ | | | $$D_{\text{eq, square}} = \frac{4 \cdot P_t^2 - \frac{\pi \cdot d_o^2}{4}}{\pi \cdot d_o}$$ Serth & Lestina (2014) |
| Equivalent Diameter for Triangular Pitch | $D_{\text{eq}}$ | | | $$D_{\text{eq, triangular}} = \frac{4 \cdot \left(0.43 \cdot P_t^2 - 0.5 \cdot \pi \cdot d_o^2\right)}{0.5 \cdot \pi \cdot d_o}$$ Serth & Lestina (2014) |
| Overall Heat Transfer Coefficient | $U_{\text{overall}}$ | | Depends on both the tube side and shell side heat transfer coefficients and fouling resistances | $$U_{\text{overall}} = \frac{1}{\frac{1}{h_s} + R_{fs} + \frac{d_o}{d_i}\left(R_{ft} + \frac{1}{h_t}\right)}$$ Serth & Lestina (2014) |
| Logarithmic mean temperature difference (LMTD) | $LMTD$ | | | $$\text{LMTD} = \frac{\Delta T_1 - \Delta T_2}{\ln\left(\frac{\Delta T_1}{\Delta T_2}\right)}$$ Serth & Lestina (2014) |
| Heat exchanger surface area | A | | | $$A = \frac{Q}{UF\text{LMTD}}$$ Serth & Lestina (2014) |
| Tube Length | $L_t$ | | Based on total heat exchanger surface area A | $$L_t = \frac{A}{\pi \cdot d_o \cdot N_t}$$ Serth & Lestina (2014) |

| Design Category | Design Variable | Types | Description | Illustrative PDE/Equation |
|---|---|---|---|---|
| Pressure drop | $\Delta P$ | Based on Kern's Method | | $$\Delta P_t = \Delta P_{\text{tube length}} + \Delta P_{\text{tube elbow}}$$ Serth & Lestina (2014) |
| | $\Delta P_{\text{tube}}$ | | Tube side pressure drop | $$\Delta P_{\text{tube}} = \frac{\rho \cdot v^2}{2} \cdot \left( \frac{L}{d_i} \cdot f + 4 \cdot n \right)$$ Serth & Lestina (2014) |
| | $\Delta P_{\text{shell}}$ | | Shell Side Pressure Drop | $$\Delta P_{\text{shell}} = \frac{\rho \cdot v^2}{2} \cdot f \cdot \frac{L \cdot D_s}{B \cdot D_e}$$ Patel & Rao (2010),Hanafi et al. (2021),Mariani et al. (2012),Hadidi et al. (2013),Abed et al. (2016) |
| | $\Delta P$ | Based on Bell-Delaware Method | , pressure drop in the central baffle spaces ($\Delta P_c$), baffle windows ($\Delta P_w$), entrance and exit baffle spaces ($\Delta P_e$) | $$\Delta P_f = \Delta P_c + \Delta P_w + \Delta P_e$$ |
| | $\Delta P$ | Based on Flow stream-analysis method | For the calculation of the pressure drop, the $\Delta P$ between two points is assumed to be the same, regardless of the paths joining these points. The pressure drop for all the streams is calculated in terms of the coefficient $n_i$ and the respective mass flow rate $\dot{m}_i$ | $$\Delta P_i = n_i \dot{m}_i^2$$ The total shell-side pressure drop, neglecting the inlet and exit nozzles, is calculated as: $$\Delta P = np\dot{m}^2 (N_b + 1)$$ where: - $n$ represents the flow coefficients, which are constants. - These coefficients are independent of the flow rate and are a function of the geometry. |
| Pumping Power | $P_{\text{pumping}}$ | | | $$P_{\text{pumping}} = \frac{1}{\eta} \left( \frac{m_{\text{tube}}}{\rho_{\text{tube}} \cdot \Delta P_{\text{tube}}} + \frac{m_{\text{shell}}}{\rho_{\text{shell}} \cdot \Delta P_{\text{shell}}} \right)$$ Serth & Lestina (2014) |
| Capital Investment Cost | $C_i$ | | where, $a_1, a_2$ and $a_3$ are constants for exchanger made with stainless steel for both shell-and-tubes | $$C_i = a_1 + a_2 \cdot S^{a_3}$$ Serth & Lestina (2014) |
| Total Discounted Operating Cost | $C_{\text{OD}}$ | | | $$C_{\text{OD}} = \sum_{k=1}^{n_y} \frac{C_o}{(1+i)^k}$$ Serth & Lestina (2014) |
| Annual Operating Cost | $C_{\text{annual}}$ | | | $$C_{\text{annual}} = P \cdot C_E \cdot H$$ Serth & Lestina (2014) |
| Total Cost | $C_{\text{total}}$ | | Total cost $C_{\text{total}}$ is taken as the objective function, which includes capital investment ($C_i$), energy cost ($C_e$), annual operating cost ($C_o$) and total discounted operating cost ($C_{\text{OD}}$) | $$C_{\text{total}} = C_i + C_o D$$ Serth & Lestina (2014) |

## C    OPTIMIZATION TECHNIQUES

The types of optimization techniques employed are listed in Table 2.

Table 2: Types of optimization technique

| NUMBER | OPTIMIZATION TECHNIQUE |
|---|---|
| 1 | PARTICLE SWARM OPTIMIZATION TECHNIQUE PATEL & RAO (2010) |
| 2 | WATER CYCLE ALGORITHM TECHNIQUE HANAFI ET AL. (2021) |
| 3 | ARTIFICIAL BEE COLONY (ABC) ALGORITHM ŞAHIN ET AL. (2011) |
| 4 | BIOGEOGRAPHY-BASED (BBO) ALGORITHM HADIDI & NAZARI (2013) |
| 5 | CUCKOO-SEARCH-ALGORITHM ASADI ET AL. (2014) |
| 6 | ANT COLONY OPTIMIZATION TECHNIQUE LAHIRI & KHALFE (2015) |
| 7 | JAYA ALGORITHM RAO & SAROJ (2017A) |
| 8 | ELECTROMAGNETISM-LIKE ALGORITHM (EM) ABED ET AL. (2016) |
| 9 | ELITIST-JAYA ALGORITHM RAO & SAROJ (2017B) |
| 10 | GRAVITATIONAL SEARCH ALGORITHM MOHANTY (2016A) |
| 11 | NSGA-II HAJABDOLLAHI ET AL. (2012) |
| 12 | MIXED-INTEGER LINEAR PROGRAMMING GONÇALVES ET AL. (2016) |
| 13 | TEACHING-LEARNING-BASED OPTIMIZATION RAO & PATEL (2013) |
| 14 | IMPROVED INTELLIGENT TUNED HARMONY SEARCH ALGORITHM TURGUT ET AL. (2014) |
| 15 | ELITIST-JAYA ALGORITHM RAO & SAROJ (2017B) |
| 16 | TSALLIS DIFFERENTIAL EVOLUTION SEGUNDO ET AL. (2016) |
| 17 | BACTERIA FORAGING ALGORITHM SAI & RAO (2019) |
| 18 | RAO ALGORITHM VENKATA RAO & PAWAR (2023) |
| 19 | GREY WOLF OPTIMISATION TECHNIQUE MAJUMDER ET AL. (2017) |
| 20 | GENETIC ALGORITHM RAVAGNANI ET AL. (2005) |
| 21 | HARMONY SEARCH ALGORITHM FESANGHARY ET AL. (2009) |
| 22 | FIREFLY ALGORITHM MOHANTY (2016B) |
| 23 | COHORT INTELLIGENCE ALGORITHM DHAVLE ET AL. (2018) |
| 24 | HEAT TRANSFER SEARCH ALGORITHM RAJA ET AL. (2017) |
| 25 | SINE COSINE ALGORITHM TURGUT (2017) |
| 26 | SYMBIOTIC ORGANISMS SEARCH TECHNIQUE MAKADIA & SANKHAVARA (2020) |
| 27 | BIO-INSPIRED ARTIFICIAL INTELLIGENCE METHODS KRZYWAŃSKI (2019) |

## D EVALUATION METRICS

In order to comprehensively assess the quality and performance of LLM and agent generated code, a multifaceted evaluation framework was devised. This framework encompasses six pivotal criteria, namely Accuracy/Correctness, Functionality, Completeness, Readability, Robustness, and Maintainability, which collectively reflect both the technical and practical dimensions of code implementation. Each of these criteria was evaluated based on predefined, specific metrics, with the code being scored on a scale of 1 to 5, where 1 denotes the lowest performance and 5 signifies the highest. This meticulous evaluation process was applied across optimization code generated by HxLLM as well as HxAgent framework on 115 articles, with multiple evaluators conducting detailed assessments of the LLM-generated code for each article.

The evaluation metrics for each criterion were carefully defined to ensure a thorough assessment. For instance, Accuracy/Correctness was evaluated by assessing the code's adherence to the logical flow and steps outlined in the reference article, with any deviations categorized as minor, moderate, or major. Functionality was measured by executing the code with test data and comparing the outputs with the expected results, while Completeness was assessed by comparing the generated code with the reference article to identify any missing or partially implemented components. Readability was evaluated based on the code's clarity, organization, and use of comments and documentation, whereas Robustness was tested by examining the code's ability to handle edge cases, errors, and unexpected inputs. Lastly, Maintainability was assessed based on factors such as modularity, code reuse, and the presence of inline comments and documentation.Further details of the evaluation process are provided in the table below (Table - 3).

## E AGENT DESCRIPTION WITH FLOW

Here are the definitions of the agents as shown in figure -7.

1. **Summary Creator Agent** ($A_S$): Reviews the given research article and generates a summary of the mathematical models using a large language model with a predefined system prompt.

2. **TF-IDF Agent** ($A_T$): Compares the generated summary with the Mathematical Model Repository and generates a similarity score, which helps determine the next steps based on a predefined threshold.

3. **Planner Agent-1** ($A_{P1}$): Creates a plan to modify the code if the similarity score suggests the model is new and not found in the repository.

4. **Designer Agent-1** ($A_{D1}$): Constructs the mathematical code based on the plan provided by Planner Agent-1.

5. **Planner Agent-2** ($A_{P2}$): If the similarity score is still below 75%, this agent repeats the planning process for a second path.

6. **Designer Agent-2** ($A_{D2}$): Builds the mathematical code based on the plan generated by Planner Agent-2.

7. **Optimization Agent** ($A_O$): Independently generates the optimization code without incorporating the mathematical model, allowing for separate analysis of the optimization code.

8. **Code Refiner Agent** ($A_C$): Reviews both the mathematical and optimization code using self-reflection principles and dataset, suggesting improvements if any gaps or missing details are found.

9. **Code Merger Agent** ($A_M$): Merges the optimization code and mathematical model code into the final existing code.

10. **Code Correction Agent** ($A_{CC}$): Iterates through the code to detect and resolve errors using the RAG (Relevant Answer Generator) techniques from the code repository.

## F REPOSITORY

HxAgent framework relies on three repositories:

Table 3: Evaluation Metrics

| Metric Name | Metric Description | Evaluation Criteria |
|---|---|---|
| Accuracy | evaluates whether the LLM-generated code correctly follows the steps, logic, and methodology described in the reference article | 1 - Completely deviates from the article's steps and methodology

2 - Several deviations from the methodology; significant errors
3 - Some deviations but mostly follows the correct logic and steps
4 - Minor deviations; almost perfectly follows the described methodology
5 - Fully adheres to the steps, logic, and methodology outlined in the article |
| Functionality | measures whether the generated code produces the correct results when executed | 1 - Code fails to execute or causes runtime errors

2 - Code executes but produces incorrect or inconsistent results
3 - Code executes with some correct results, but issues or inconsistencies remain
4 - Code executes correctly with minor inaccuracies or edge cases
5 - Code executes perfectly, and all results match the expected outputs |
| Completeness | assessed whether the generated code covered all aspects of the implementation described in the reference article | 1 - Major components or steps are missing

2 - Several parts of the implementation are absent or incomplete
3 - Some minor components are missing or only partially implemented
4 - Almost complete; only minor details are missing
5 - Fully complete; covers all aspects as described in the article |
| Readability | Readability refers to how well-structured and easy to understand the code is | 1 - Very unclear, difficult to follow

2 - Somewhat unclear, poor structure
3 - Understandable, but with minor readability issues
4 - Mostly clear, well-organized
5 - Very clear, easy to understand, well-structured, with excellent comments and documentation |
| Robustness | Robustness evaluates how well the code handles edge cases, errors, and unexpected inputs | 1 - No error handling, fails on most edge cases

2 - Limited error handling, fails on many edge cases
3 - Some error handling, covers basic edge cases
4 - Good error handling, manages most edge cases well
5 - Excellent error handling, robust against all edge cases and unexpected inputs |
| Maintainability | Maintainability assesses how easy it is to update and modify the generated code in the future | 1 - Very hard to maintain or update, no modularity
2 - Somewhat hard to maintain, minimal modularity
3 - Reasonably maintainable with some effort, some modularity
4 - Mostly easy to maintain and update, good modularity
5 - Very easy to maintain, well-structured with excellent modularity |

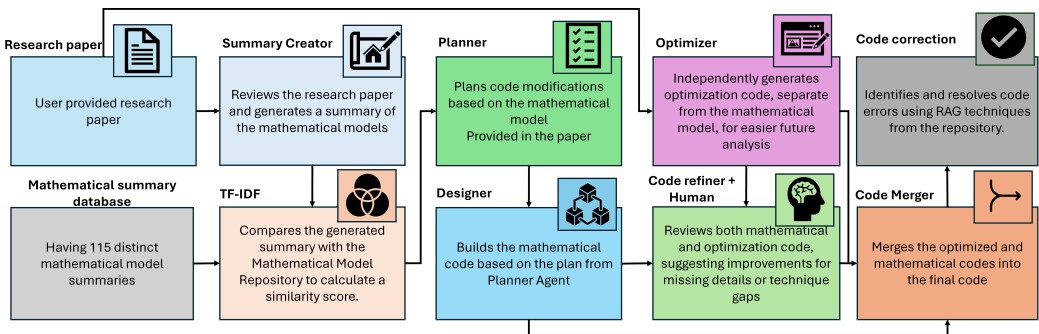

Figure 7: Agentic Framework

1. Mathematical model and code repository ($M$)
2. Code error repository ($CE$)
3. Self reflection repository ($SR$)

Here are the details of each repositories.

## F.1 MATHEMATICAL MODEL REPOSITORY :

The Mathematical Model Repository is a CSV-based database containing over 115 distinct entries, each organized into three columns: ID, Summary (S), and Code (C). The Summary column includes brief overviews of research articles, which were derived using structured prompts. For each mathematical model, an expert-developed Python script was written to evaluate the associated equations. These scripts were processed through an error-correction framework to ensure their accuracy and reliability before being added to the Code section of the repository.

## F.2 CODE ERROR REPOSITORY:

The Code Error Repository is a specialized collection documenting errors encountered by the LLM during the generation of optimization code. These errors are typically complex in nature and represent challenges in the translation of theoretical models into executable code. Each error entry in the repository includes a description of the error (RAG-E) and the corresponding solution (RAG-S), providing a valuable resource for debugging and improving the LLM's code-generation capabilities.

## F.3 SELF-REFLECTION INFORMATION DATA REPOSITORY:

The Self-Reflection Information Dataset consists of a collection of Word documents that include a variety of mathematical models along with their associated equations. Additionally, this dataset contains a comprehensive list of optimization algorithms and their respective processes, which are essential for refining and optimizing heat exchanger designs. This dataset serves as the foundation for the HxAgent framework, facilitating the iterative improvement of generated solutions by combining LLM output with human expertise.

## G ILLUSTRATIVE EXAMPLES COMPARING THE PERFORMANCE OF HXAGENT AND HXLLM FRAMEWORK

In this section, we compare the code generation capabilities of the HxLLM framework and the HxAgent framework. This comparison reveals significant advantages of the HxAgent framework over the HxLLM framework. The HxAgent framework consistently ensures the correct selection of optimization algorithms (see G.2), enhances functionality and robustness (see G.3), ensures accurate variable selection (see G.1), and produces more complete and usable code. While the HxLLM framework is functional, it often deviates from the original article's specifications, resulting in errors

and incomplete solutions. In contrast, the HxAgent framework leverages a self-reflection mechanism and specialized agents to refine and improve the generated code, leading to more accurate, robust, and complete results.

## G.1 CORRECT SELECTION OF VARIABLES:

The HxAgent framework accurately defines key variables, such as tube outside diameter and shell inside diameter, in alignment with the original article 'Adaptive Range Genetic Algorithm: A Hybrid Optimization Approach' by Iyer et al. (2019). In contrast, the HxLLM framework fails to do so when generating optimization code. HxLLM framework does not correctly define important variables like tube outside diameter, shell inside diameter, and baffle spacing, resulting in discrepancies with the original specifications. On the other hand, HxAgent framework ensures proper identification and definition of these variables, maintaining consistency with the original article (see 4).

Table 4: Correct Selection of Variables

| Variable selection by HxLLM Framework | Variable selection by HxAgent framework |
|---|---|

```
# Define the optimization algorithm
def adaptive_range_genetic_algorithm(objective_function,
bounds, population_size=100, generations=100, mutation_
rate=0.1, reduction_factor=0.96):
    population = [[random.uniform(bounds[i][0], bounds
    [i][1]) for i in range(len(bounds))] for _ in range
    (population_size)]
    for _ in range(generations):
        fitness = [objective_function(*individual) for
        individual in population]
        parents = [population[i] for i in np.argsort
        (fitness) [:int(population_size/2)]]
        offspring = []
        for _in range(population_size - len(parents)):
            parent1, parent2 = random.sample(parents,2)
            child = [(parent1[i] + parent2[i]) / 2 for
            i in range (len(parent1))]
            if random.random() < mutation_rate:
                child = [child[i] + random.uniform
                (-bounds[i][1] * reduction_factor,
                bounds[i][1] * reduction_factor)
                for i in range(len(child))]
            offspring.append(child)
        population = parents + offspring
    return population[0]

# Define the bounds for the design variables
bounds =[(1.0, 2.0),(0.05, 0.1),(4,6),(0.5, 1.0),(4,6)]

# Run the optimization algorithm
optimized_design = adaptive_range_genetic_algorithm
(linking_function, bounds)

# Print the optimized design
print("Optimized design:", optimized_design)

#Evaluate the objective function at optimized design
optimized_cost = linking_function(*optimized_design)
print("Optimized cost:", optimized_cost)
```

```
# Define the optimization algorithm
def optimization_algorithm(Ds,do,B,mt,rho_t,rho_s, ...):
    # Initialize population
    population = [{'Ds': random.uniform(0.1, 1.5), 'do':
    random. uniform(0.015, 0.051), 'B': random.uniform
    (0.05, 0.5), 'fitness': float('inf')} for _
    in range(100)]
    # Optimization parameters
    Cr, Mr, max_iterations = 0.5, 0.1, 100
    for _ in range(max_iterations):
        # Evaluate fitness (simplified)
        for individual in population:
            hs, ht = Shell_Side_heat_transfer
            _coefficient(...),
            Tube_side_heat_transfer
            _coefficient(...)
            U = 1 / (hs + ...)
            A = (mt * Cp_c * ...) / (U * ...)
            Ctot = total_cost(..., ...)  # Simplified
            cost function
            individual['fitness'] = Ctot
        # Select fittest individual
        fittest_individual = min(population, key=lambda
        individual:  individual['fitness'])
        # Crossover and mutation
        for individual in population:
            if random.random() < Cr:
                parent1, parent2 = random.choice
                (population), random.choice(population)
                individual.update({'Ds':(parent1['Ds'] +
                parent2['Ds'])/ 2,'do':(parent1['do'] +
                parent2['do']) / 2, 'B':
                (parent1['B'] + parent2['B']) / 2})
            if random.random() < Mr:
                individual['Ds']+=random.uniform
                (-0.1,0.1)
                individual['do']+=random.uniform
                (-0.01,0.01)
                individual['B']+=random.uniform
                (-0.01,0.01)
        population = population[:100]
    return fittest_individual

# Run the algorithm
Ds, do, B = 1.5, 0.051, 0.5
fittest_individual =optimization_algorithm(Ds,do,B,...)

# Display fittest result
print('Fittest individual:', fittest_individual)
```

## G.2 CORRECT SELECTION OF OPTIMIZATION ALGORITHM:

The HxAgent framework correctly identifies and applies the MOPSO algorithm, whereas the HxLLM framework incorrectly uses SciPy for optimization. The HxLLM framework initially selects the wrong optimization algorithm, opting for SciPy instead of the intended MOPSO (Multi-Objective Particle Swarm Optimization), deviating from the method outlined by Ghanei et al. (2014) in the

original article. In contrast, the HxAgent framework accurately identifies and applies the MOPSO algorithm after interacting with specialized agents and incorporating self-reflection, ensuring precise code generation (see 5).

Table 5: Correct Selection of Optimization Algorithm

| Optimization code by HxLLM Framework | Optimization code by HxAgent framework |
|---|---|
| <pre>import math
import numpy as np
from scipy.optimize import minimize


# ...

# Optimization function
def optimize_heat_exchanger_performance():
    # Define the bounds for the design variables
    bounds = [
        (0.0112, 0.0153),  # Tube diameter
        (1.25, 2),  # Tube pitch
        (3, 8),  # Tube length
        (100, 600),  # Number of tubes
        (0.19, 0.32),  # Baffle cut
        (0.2, 1.4),  # Baffle spacing
        (3, 12),  # Number of baffles
    ]

    # Define the objective function
    def objective(x):
        di, pt, L, N_t, BC, bs, N = x
        performance = calculate_heat_exchanger_
        performance(
            di,pt,
            .........

            0.9,  # Efficiency of the heat exchanger
        )
        return -performance["effectiveness"] +
        performance["C_tot"]

    # Perform the optimization
    result = minimize(objective, [0.013, 1.5, 5, 300,
    0.25,  0.6, 6], method="SLSQP", bounds=bounds)

    return result.x

# Perform the optimization
optimized_design = optimize_heat_exchanger_performance()
print(optimized_design)</pre> | <pre>def MOPSO(objective_function, num_particles,
num_iterations, num_dimensions, bounds):
    # Initialize particles
    particles = []
    for _ in range(num_particles):
        particle = []
        .......
    # Initialize velocities
    velocities = []
    for _ in range(num_particles):
        velocity = []
        for _ in range(num_dimensions):
            velocity.append(random.uniform(-1, 1))
    .........
    # Initialize personal best positions
    personal_best_positions = particles[:]

    # Initialize global best position
    global_best_position = particles[0]

    # Run algorithm
    for _ in range(num_iterations):
        for i in range(num_particles):
            # Calculate fitness
            fitness = objective_function(particles[i])

            # Update personal best position

            # Update global best position
            if fitness < objective_function(global
            _best_position):
                global_best_position = particles[i]

            # Update velocity
            velocities[i] = [velocities[i][j] + 0.5 *
            random.random() * (personal_best_positions
            [i][j]
            .........

            # Update position
            particles[i] = [particles[i][j] + ......
# Define objective function
def objective_function(x):
    k_t, di, Re_t, Pr_t, h_id, J_c, J_l, J_b,
    J_s, J_r,... =return effectiveness, C_tot

# Define bounds for each dimension
bounds = [(10, 100), (0.01, 0.1), (1000, 10000),
.......... (1000, 10000)]

# Run MOPSO algorithm
pareto_front = MOPSO(objective_function, 100, 100, 23,
bounds)

# Print Pareto front
for solution in pareto_front:
    print(solution)</pre> |

## G.3 IMPROVING FUNCTIONALITY AND ROBUSTNESS:

The HxAgent framework generates functional, executable code, while the HxLLM framework sometimes produces code that remains non-runnable even after correction attempts. In the case of the article Roy & Majumder (2019) on economic optimization and energy analysis in shell and tube heat exchangers, the HxLLM framework generates code that is not runnable, even after undergoing a correction loop. The HxAgent framework, however, successfully generates executable code, demonstrating its superior ability to create functional and robust solutions (see 6). Explanation of changes:

1. Modified the krill_herd_motion function to perform element-wise multiplication between N_max and n * N_old.

2. Updated the `krill_herd_optimization_algorithm` function to incorporate the new `krill_herd_motion` function and perform element-wise operations.

# H    EXAMPLE ILLUSTRATING THE PROCESS OF HxAGENT

We have applied the HxAgent framework to 115 articles, and in this subsection, we highlight one example, by Rao et al. (Patel & Rao, 2010) from Group 1, which focuses on minimization of cost based on geometrical parameters using a mathematical model and optimization of Heat Exchanger design.

## H.1    MATHEMATICAL MODEL EXTRACTION

The first step involves the summary creator agent($A_S$). Scientific articles (P) containing the mathematical model for heat exchanger design is provided as context to the summary creator agent, which generates a summary of all the mathematical model equations present in the article. The output of this step is shown below.

```
Here is the list of unique equations used to construct the model:
1. **Particle Velocity Update**: Calculates the new velocity of a particle based on its previous velocity, the
best location it has  achieved, and the global best location, using acceleration constants and random terms.
2. **Particle Position Update**: Updates the position of a particle in the solution hyper space based on its
new velocity.
..................
25. **Exchanger Length (L)**: Calculates the exchanger length based on the heat exchanger surface area, tube
diameter, and number of  tubes.
```

## H.2    BASE CODE RETRIEVAL FOR MATHEMATICAL MODEL

The extracted model was compared to similar models in the mathematical model repository using the TF-IDF agent($A_T$) . Here in this example the framework achieved a similarity score of 0.61 (below the threshold of 0.75), indicating that the generated code was based on a non-relevant reference model. Based on the closest match index, the most similar mathematical code was retrieved and sent to the next step as a base code for the planner agent($A_P$) . The framework then determines the appropriate path based on the score and directs it to Path-2, involving both Planner Agent-2($A_{P2}$) and Designer Agent-2($A_{D2}$) .

```
Code example:
0.6089355563042992
25

"Code for reference":
import math
def total_annual_cost_PFHE(Cin, Cop):
    """Equation (13)"""
    return a * Cin + Cop
..............
# Function for all the rest equations of  mathematical model

def optimize_heat_exchanger(NTU, C, CA, Atot, n, kel, s, DPc, Vc, gc, DPh, Vh, gh, r, q, At, ny,  i, Cop, gp,
DPt, Vt, DPs, Vs, Z1_max, Z2_min, w1, R1, hh, hc):
    # Calculate effectiveness of PFHE
    eff_PFHE = effectiveness_PFHE(NTU, C)
    # Calculate capital and operating costs of PFHE
    Cin_PFHE = capital_cost_PFHE(CA, Atot, n)
    a = annual_cost_coefficient(r, q)
    Cop_PFHE = operating_cost_PFHE(kel, s, DPc,  Vc, gc, DPh, Vh, gh)
    # Calculate total annual cost of PFHE
    total_cost_PFHE = total_annual_cost_PFHE(Cin_PFHE, Cop_PFHE)
..............
# Example usage
NTU = 2.5
C = 0.8
................
eff_PFHE, total_cost_PFHE, eff_STHE, total_cost_STHE,  Z = optimize_heat_exchanger(NTU, C, CA, Atot, n, kel,
s, DPc, Vc, gc, DPh, Vh, gh, r, q, At, ny, i, Cop, gp,  DPt, Vt, DPs, Vs, Z1_max, Z2_min, w1, R1, hh, hc)
```

## H.3    CONSTRUCTING THE MATHEMATICAL MODEL

The planner agent - 2($A_{P2}$), with the help of the base code, first identifies the unique equations (see code example A) used in the mathematical model. It then creates a plan for implementing these equations to build an objective function, which will be used in the optimization process. The plan is then passed to Designer Agent-2($A_{D2}$) for further action.Following the planning phase, the Designer

Agent takes on the task of constructing the overall objective function, as described in the article. It includes all the distinct equations outlined in the article by using the base code as a reference for constructing an objective function. The agent's output is presented in equation (see code example B).

```
Code Example - A:
---DECISION: THERE IS NO SIMILAR article FOUND IN THE
DATABASE---
---STEP-1:IDENTIFYING THE UNIQUE EQUATIONS---
I'll start by creating the functions for the mathematical models described in the article. Here's the initial
code:
```python
import math

def Prandtl_number(mew, Cp, Kt):
    Pr = (mew * Cp) / Kt
    return Pr

def calculate_fs(b0, Re):
    fs = 2 * b0 * Re**(-0.15)
    return fs

def Number_of_tubes(C, n1, Ds, d0):
    Nt = C * ((Ds / d0)**n1)
    return Nt
..............
# Function for all the rest equations of mathematical model

def calculate_DPs(ms, mew_s, rho_s, pitch, d0, L, B, Ds):
    St = 1.25 * d0
    As = Cross_section_area(Ds, B, d0)
    de = Shell_hydaulic_diameter(pitch, d0)
    Re_s = Reynolds_number_shell(ms, de, As, mew_s)
    b0 = 0.72
    fs = calculate_fs(b0, Re_s)
    vs = Flow_velocity_shell(ms, rho_s, As)
    DPs = fs * (rho_s * (vs**2) / 2) * (L / B) * (Ds / de)
    return DPs
```
Please let me know if this is correct and if I should proceed with the rest of the code. Also, I noticed that
the article mentions the Particle Swarm  Optimization (PSO) algorithm, but it doesn't provide the specific
implementation details. If you'd like me to implement the PSO  algorithm, please provide more information on
how you'd like it to be implemented. Please provide the next prompt to guide me through the process.

Code Example - B:
import math
import numpy as np

# Constants
a1 = 8000
a2 = 259.2
.........
i = 0.1  # Annual discount rate (%)

def calculate_ht(mt, rho_t, n, Nt, di, L, mew_t, Cp_t, kt):

    vt = mt / ((math.pi / 4) * (di**2) * rho_t * n * Nt)
    Ret = (rho_t * vt * di) / mew_t
    Prt = (mew_t * Cp_t) / kt

    if Ret < 2300:
        ht = (kt / di) * (3.657 + (0.0677 * (Ret * Prt * (di / L)
            **1.33)**0.33) / (1 + (0.1 * Prt * (Ret * (di / L)))**0.3))
................
def calculate_Ctot(Ci, Cod):

    Ctot = Ci + Cod
    return Ctot

def calculate_total_cost(
    C, n1, Ds, mt, rho_t, rho_s, n, mew_t, L, Kt, pitch, d0, B, ms, mew_s, Ks, Rfs, Rft, Th_i, Th_o, Tc_i,
    Tc_o, Cp_h, Cp_c, etta, ny, H, Ce, i):
    p = 4
    di = 0.8 * d0
    hs = Shell_Side_heat_transfer_coefficient(Kt, pitch, d0, Ds, B,
    ms, mew_s, Cp_h, Ks)
    ht = Tube_side_heat_transfer_coefficient(C, n1, Ds, mt, rho_t,
    n, mew_t, Cp_c, d0, L, Kt)
    U = 1 / ((1 / hs) + Rfs + (d0 / di) * (Rft + (1 / ht)))
    lmtd = LMTD(Th_i, Th_o, Tc_i, Tc_o)
    ....
    C_tot = total_cost(C_i, C_od)
    return C_tot

# Example usage
C = 0.158
...
Ce = 0.12
i = 10

Cost = calculate_total_cost(
```

```
    C, n1, Ds, mt, rho_t, rho_s, n, mew_t, L, Kt, pitch, d0, B, ms, mew_s, Ks, Rfs, Rft, Th_i, Th_o, Tc_i,
    Tc_o, Cp_h, Cp_c, etta, ny, H, Ce, i)
print(f"Total cost: {Cost}")
```

## H.4 CONSTRUCTING THE OPTIMIZATION ALGORITHM

To further refine the code, the Optimization Agent ($A_O$) focuses exclusively on optimization algorithm code generation. As shown in output (see code example - C), the Optimizer Agent first selects the type of optimization algorithm, followed by the objective function, constraints, and variables. It then creates a dummy objective function based on these elements and attempts the optimization process.

```
Code Example - C:
**Optimization Algorithm: Particle Swarm Optimization (PSO)**

The research article describes a Particle Swarm Optimization (PSO) algorithm for optimizing the design of a
heat exchanger. The objective function is to minimize the total cost, which includes capital investment, energy
cost, and annual operating cost.
**Optimization Problem**
* Objective function: Minimize the total cost `Ctot`
* Constraints:
+ Shell internal diameter `Ds` between 0.1 m and 1.5 m
+ Outer tube diameter `do` between 0.015 m and 0.051 m
+ Baffle spacing `B` between 0.05 m and 0.5 m
* Variables:
+ `Ds`: Shell internal diameter
+ `do`: Outer tube diameter
+ `B`: Baffle spacing

**PSO Algorithm**

The PSO algorithm can be implemented using the following steps:

1. Initialize the population of particles with random positions and velocities.
2. Evaluate the objective function for each particle.
3. Update the velocity and position of each particle using the following equations:
* `V_i+1 = w * V_i + c1 * r1 * (pBest_i - X_i) + c2 * r2 *
(gBest - X_i)`
* `X_i+1 = X_i + V_i+1`
4. Update the `pBest` and `gBest` values.
5. Repeat steps 2-4 until the maximum number of iterations is reached.

**Code Implementation**

Here is a Python implementation of the PSO algorithm:
```python
import numpy as np

def objective_function(X):
    # Calculate the total cost
    Ds, do, B = X
    # Calculate the heat exchanger surface area
    A = calculate_surface_area(Ds, do, B)
    # Calculate the capital investment
    Ci = calculate_capital_investment(A)
    # Calculate the energy cost
    Co = calculate_energy_cost(Ds, do, B)
    # Calculate the total cost
    Ctot = Ci + Co
    return Ctot

.............
def pso(objective_function, bounds, num_particles,
num_iterations):
    # Initialize the particles
    particles = np.random.uniform(bounds[0], bounds[1],
    size=(num_particles, len(bounds[0])))
    velocities = np.random.uniform(-1, 1, size=(num_particles, len(bounds[0])))
    pBest = particles
    gBest = np.mean(particles, axis=0)
    # Iterate through the generations
    for i in range(num_iterations):
        # Evaluate the objective function for each particle
        fitness = np.array([objective_function(p) for p in particles])
        # Update the pBest and gBest values
        for j in range(num_particles):
            if fitness[j] < objective_function(pBest[j]):
                pBest[j] = particles[j]
        gBest = np.mean(pBest, axis=0)
        # Update the velocities and positions of the particles
        for j in range(num_particles):
            velocities[j] = 0.729 * velocities[j] + 1.494 * np.random.uniform(0, 1) * (pBest[j] -
            particles[j]) + 1.494 * np.random.uniform(0, 1) * (gBest -
            particles[j])
            particles[j] = particles[j] + velocities[j]
    return gBest

# Define the bounds for the variables
bounds = [(0.1, 1.5), (0.015, 0.051), (0.05, 0.5)]
```

```
# Define the number of particles and iterations
num_particles = 100
num_iterations = 100

# Run the PSO algorithm
gBest = pso(objective_function, bounds, num_particles, num_iterations)

print("Optimal solution:", gBest)
print("Optimal cost:", objective_function(gBest))
```
Note that this is a simplified implementation of the PSO algorithm and may not include all the features and
nuances of the original algorithm. Additionally, the objective function and constraints are based on the
equations provided in the research article and may not be accurate or up-to-date.

## H.5   APPLYING SELF-REFLECTION

In this stage, we utilize a self-reflection agent ($A_C$) alongside a dedicated dataset to evaluate whether
the generated code meets the required criteria. As described in (see code example - D) The interaction
focuses on potential issues within the mathematical model, objective function, and optimization
algorithm. This evaluation process allows the system to identify areas that may require improvement.
A human-in-the-loop (HITL) then reviews and approves the outcome of this reflection, ensuring
the validity of the suggestion. If the suggestion appears to be incorrect, the system will perform
self-reflection again until the output meets the requirements. However, this step is optional for the
human reviewer as well.

```
Code Example - D:
**Review of Mathematical Model Code**
The mathematical model code provided appears to be a comprehensive implementation of the heat exchanger design
equations and formulas presented in the research article. The code includes functions for calculating the heat
transfer coefficients, overall heat transfer coefficient, heat exchanger area, investment cost, operating cost,
and total cost.

However, upon closer inspection, there are a few areas where the code could be improved:

1.  **Variable naming**: Some variable names, such as 'C', 'n1', 'Ds', 'mt', 'rho_t', 'rho_s', 'n', 'mew_t',
'L', 'Kt', 'pitch', 'd0', 'B', 'ms', 'mew_s', 'Ks', 'Rfs', 'Rft', 'Th_i', 'Th_o', 'Tc_i','Tc_o', 'Cp_h', 'Cp_c'
, 'etta','ny', 'H', 'Ce', and 'i', are not very descriptive. It would be better to use more descriptive names
to improve code readability.
2.  **Function organization**: The code has many functions, but they are not organized in a logical manner.
3.**Error handling**: The code does not include any error handling mechanisms. It would be a good idea to add
try except blocks to handle  potential errors, such as division by zero or invalid input values.
4.  Code comments: The code could benefit from additional comments to explain the purpose of each function and
how it is used in the overall calculation.

**Review of Optimization Algorithm Code** There are a few areas where the code could be improved:

1.  **Objective function**: The objective function is not clearly defined in the code. It would be better to
define the objective function explicitly,so that it is clear what the optimization algorithm is trying to
minimize.
2.  **Optimization algorithm**: The code uses a simple PSO implementation, but the logic for selecting the best
solution is not clearly defined.
3.  **Code comments**: The code could benefit from additional comments to explain the purpose of each section
and how it is used in the overall optimization process.

**Check for Missing or Incomplete Information**Upon reviewing the code, it appears that there are a few areas
where information is missing or incomplete:

1.  **Heat exchanger design specifications**: The code does not include any information about the heat exchanger
design specifications, such as the type of heat exchanger, the fluids used, or the operating conditions.
2.  **Optimization algorithm settings**: The code does not include any information about the optimization
algorithm settings, such as population size, the number of generations, or the crossover and mutation rates.
3.Objective function definition: The code does not include a clear definition of the objective function, which
is used to evaluate the fitness of each solution.

**Evaluate Logic and Consistency** The code appears to be logically consistent, but there are a few areas where
the logic could be improved:

1.  **Heat transfer coefficient calculations**: The code uses different equations to calculate the heat transfer
coefficients, depending on the flow regime. However, the logic for selecting the correct equation is not clearly
defined.
2.  **Optimization algorithm**: The code uses a simple PSO implementation,but the logic for selecting the best
solution is not clearly defined.
3 **Error handling**: The code does not include any error handling mechanisms, which could lead to errors or
inconsistencies in the results.

**Provide Feedback and Recommendations**
Based on the review of the code, the following feedback and recommendations are provided:
1.  **Improve variable naming**: ....
2.  **Organize functions**: ....
3.  **Add error handling**: ....
4.  **Define objective function**: ....
5.  **Include heat exchanger design specifications**: ....
6.  **Define optimization algorithm settings**: ....

By addressing these areas, the code can be improved to provide a more accurate and efficient solution to the heat
exchanger design problem.
```

## H.6    FINAL CODE CORRECTION AND GENERATION

Next, the Code Merger Agent ($A_M$) combines the mathematical model code with the optimization algorithm code, incorporating any feedback or adjustments suggested during the self-reflection process. Then the Code Correction Agent($A_{CC}$) conducts a thorough validation check to ensure the generated code is accurate and error-free before it is finalized. Simple errors are corrected using LLM prompts, while known and complex issues are addressed through a RAG-based approach, resulting in a clean, error-free code. At this stage, a human reviewer is involved to assess the quality of the code. If the quality does not meet expectations, the self-reflection agent again provides suggestions, and the code merger agent rewrites the code. The output of code merger agent is presented below (see code example - E).

```
Code Example - E:
import math
import random as rd

# Constants
w = 0.9  # Inertia weight
c1 = 2  # Cognitive parameter
c2 = 2  # Social parameter
Vmax = 1  # Maximum velocity
num_particles = 50  # Number of particles
num_generations = 100  # Number of generations

# Define the heat exchanger model functions
def Prandtl_number(mew, Cp, Kt):
    Pr = (mew * Cp) / Kt
    return Pr
#Rest of the model ....

# Define the objective function
def objective_function(x):
    Ds, do, B = x
    hs = Shell_Side_heat_transfer_coefficient(0.13, 4, do, Ds, B, 5.52,0.0004, 2.47, 0.13)
    ht = Tube_side_heat_transfer_coefficient(0.158, 2.263, Ds, 18.80, 995, 4, 0.00358, 2.05, do, 4.88, 0.13)
    U = 1 / ((1 / hs) + 0.00061 + (do / (do * 0.8)) * (0.00061 +
    (1 / ht)))
    ....
    return Ctot

# Define the PSO algorithm
def pso(bounds, num_particles, num_generations):
    particles = []
    for _ in range(num_particles):
        particle = {
            'position': [rd.uniform(bounds[0], bounds[1]) for _ in range(3)],
            'velocity': [rd.uniform(-Vmax, Vmax) for _ in range(3)],
            'best_position': [rd.uniform(bounds[0], bounds[1]) for _ in range(3)],
            'best_cost': float('inf')
        }
        particles.append(particle)

    global_best_position = [rd.uniform(bounds[0], bounds[1]) for _ in range(3)]
    global_best_cost = float('inf')
....
    return global_best_position, global_best_cost

# Run the PSO algorithm
bounds = [0.1, 1.5]
best_position, best_cost = pso(bounds, num_particles,
num_generations)
print('Best position:', best_position)
print('Best cost:', best_cost)

Code Example - F
...PRESENTING GENERATED CODE TO HUMAN VALIDATOR...
Is the code valid? (Yes/No):Only answer in Yes/No
Suggestion given by human validator...
Yes
---ASSESS GENERATED CODE---
---DECISION: CODES ARE RELEVANT TO QUESTION, CONTINUE---

Code has no error
Best position: [1.5, 0.1, 1.5]
Best cost: 9326.672051995894
```

## I    DETAILS OF EVALUATION PROCESS

In this section, we discuss how we evaluated the code for 115 research articles generated by HxLLM and HxAgent framework using and the six predefined evaluation metrics.

## I.1 ACCURACY/CORRECTNESS

We demonstrated our evaluation (see details in Table - 7) approach by assessing the code generated for Papers "A" (Selbaş et al., 2006), "B" (Abbasian Arani & Moradi, 2019), and "C" (University & Shaik, 2007). The paper-A introduces a new design approach for shell-and-tube heat exchangers using genetic algorithms from an economic perspective, and the generated code accurately implements the genetic algorithm as described. It earned a score of 4, exhibits only minor deviations and closely follows the methods outlined in the paper. Paper C, which scored 3 out of 5, showed some deviations from the original content. This paper explores differential evolution strategies for the optimal design of shell-and-tube heat exchangers, involving several complex steps. While the generated code generally follows the methods, there are areas, particularly in the classification steps, where improvements could be made. Finally, Paper B discusses shell-and-tube heat exchanger optimization using a new baffle and tube configuration, leveraging CFD tools for optimization. Since CFD is outside the scope of our framework, the framework used PSO techniques instead, which were not mentioned in the paper. As a result, the code received a score of 2 out of 5, reflecting several deviations from the described methodology.

## I.2 FUNCTIONALITY

Test cases and scenarios designed in the original paper were used to validate the functionality of the code. Additional edge cases were also considered to evaluate the behavior of the code under less common conditions. The goal was to identify not only correctness in output but also reliability and consistency during execution. As shown in the table (8) below, we have taken three papers as examples: Paper "A", Paper "B", and Paper "C". For Paper A, after the code is generated, it fails to execute and produces a run time error with the following message:

```
return (1.82 * math.log10(Re) - 1.64)**-2
TypeError: must be real number, not complex
```

As a result, it received a score of 1 out of 5. For Paper B, the code executes correctly, yielding the following output:

```
Best position: [1.5, 0.015, 0.5]
Best cost: 4878150935878.0
```

Since the code executes as expected, it received a score of 4 out of 5. Lastly, for Paper C, the code produces the output:

```
Best country: None
```

This indicates that while the code runs, some correct results are produced, but there are still issues or inconsistencies. Therefore, it received a score of 3 out of 5.

## I.3 COMPLETENESS

The Completeness criterion assessed whether the generated code covered all aspects of the implementation described in the reference paper. Missing or incomplete implementations can significantly reduce the overall effectiveness of the code, limiting its real-world applicability. In cases where portions of the code were incomplete, the evaluators identified whether those omissions were crucial to the code's functionality or whether they had a minimal impact on the overall outcome. As shown in the table (9) below, we have taken three papers as examples: Paper "A" (Asadi et al., 2014), Paper "B"(Selbaş et al., 2006), and Paper "C" (Lahiri & Khalfe, 2015). Paper C discusses the use of Hybrid Particle Swarm Optimization (PSO) and Ant Colony Optimization (ACO) techniques for the optimal design of shell-and-tube heat exchangers. However, the final code primarily focuses on the PSO algorithm, with only minimal implementation of ACO techniques. As a result, it received a score of 3 out of 5, as it falls under the category of "Some minor components are missing or only partially implemented." Paper B presents a new design approach for shell-and-tube heat exchangers using genetic algorithms from an economic perspective. The code perfectly implements the genetic algorithm and includes almost all the equations mentioned in the paper's mathematical model. Therefore, it falls under the category of "Fully complete; covers all aspects as described in the paper," and received

a score of 5 out of 5. Finally, Paper A discusses the economic optimization design of shell-and-tube heat exchangers using a cuckoo-search algorithm. The code provided in the table implements the cuckoo-search algorithm almost perfectly as described in the paper. Since only some minor details are missing from math model as well as from optimization model, it received a score of 4 out of 5, falling under the category of "Almost complete; only minor details are missing."

## I.4 READABILITY AND MAINTAINABILITY

Readability refers to how well-structured and easy to understand the code is. Well-written code that is easy to follow is essential for future modifications, collaboration, and debugging. This criterion also emphasizes how the code is formatted and whether proper variable names, comments, and documentation are used. Specific elements like the use of descriptive variable names, logical indentation, and comments explaining complex code sections were key factors in determining readability. Code that was easy to navigate and modify in the future was rated higher. Maintainability assesses how easy it is to update and modify the generated code in the future. Maintainable code is modular, well-commented, and follows good coding practices, making it easier for others to enhance or debug. Code that was modular, with functions and methods that could be easily updated or reused, scored higher in terms of maintainability. Additionally, clear documentation and inline comments that explained the purpose and usage of different sections of the code were considered positive aspects. For this comparison, we selected Paper "A" (Asadi et al., 2014) and Paper "B" (Jamil et al., 2020). Paper B is well-organized, with clearly defined variables and functions. The code follows a logical structure: it begins with the mathematical model, followed by the objective function, and concludes with the optimization algorithm. This organization contributes to a rating of 4 out of 5, categorized as "Mostly clear, well-organized." In contrast, Paper A lacks clear descriptions of the variables and could benefit from improved structure, particularly by separating the optimization and objective functions. While the code remains understandable and accurate, the lack of modularity reduces its clarity. As a result, it receives a score of 3 out of 5, categorized as "Understandable, but with minor organizational and readability issues(see details in Table - 10)."

## I.5 ROBUSTNESS

Robustness evaluates how well the code handles edge cases, errors, and unexpected inputs. Code that is robust can maintain functionality and provide meaningful error messages when facing edge conditions, which is crucial for its real-world deployment. Code that failed to handle specific edge cases or crashed upon receiving unexpected inputs was rated lower. On the other hand, code that effectively managed exceptions and returned helpful error messages scored higher. For this example (details in Table - 11), we have selected Papers "A" (Lahiri & Khalfe, 2015), "B" (Şahin et al., 2011), and "C" (University & Shaik, 2007). For Paper C, it works for almost all cases. The output is as follows:

```
Best position: [1.5, 0.051, 0.5]
Best cost: 18239.5649722467
```

Therefore, it has good error handling, manages most edge cases well, and received a rating of 4 out of 5. For Paper B, the output only appears after defining the following function:

```
def LMTD(Th_i, Th_o, Tc_i, Tc_o):
    deltaT1 = Th_i - Tc_o
    deltaT2 = Th_o - Tc_i
    lmtd = (deltaT1 / deltaT2)
    return lmtd
```

This simplifies lmtd to deltaT1 / deltaT2. However, in the case of the formula lmtd = (deltaT1 - deltaT2) / math.log(deltaT1 / deltaT2), a mathematical domain error occurs. As a result, this paper falls under the category of "Some error handling, covers basic edge cases," and received a rating of 3 out of 5. For Paper A, it fails to handle errors properly and struggles with most edge cases. Consequently, it received a rating of 2 out of 5.

## J    LIMITATIONS AND FUTURE WORK

The evaluation was limited to predefined criteria, including accuracy, functionality, completeness, readability, robustness, maintainability, and overall performance. Other important aspects, such as computational efficiency and scalability, were not thoroughly examined. Additionally, while the challenges faced by Groups 2, 4, 5, and 6 were discussed, the specific methodologies and data limitations impacting these groups were not explored in detail. The agentic framework did not incorporate external simulation tools, such as CFD tools, which could have supported the simulation process. Furthermore, the study lacked a structured approach, predefined datasets, and established methodologies for situations involving heat exchangers within complex networks.

Future research should focus on broadening the analysis by integrating additional frameworks and more varied datasets to better understand the strengths and limitations of the HxAgent framework. A deeper examination of computational efficiency and scalability could also provide valuable insights for real-world applications. Moreover, it would be beneficial to investigate the incorporation of real-world data and more rigorous validation techniques to address the data handling and model development challenges identified in Groups 2, 4, 5, and 6.

Table 6: Improving Functionality and Robustness

| Code by HxLLM Framework | Code by HxAgent framework |
|---|---|

```
Code {
import math
import random

def Prandtl_number(mew, Cp, Kt):
    ...

def krill_herd_motion(N_max,n,N_old,i_local,i_target):
    ...
    return N_max * (n * N_old + i_local + i_target)

def krill_herd_foraging(F_i, m, F_food, F_best):
    ...
    return F_i * (m * F_food + F_best)

def krill_herd_diffusion(P_max, lambda_):
    ...
    return P_max * lambda_

def krill_herd_optimization_algorithm(C, n1, Ds, mt,
rho_t,
rho_s, n, mew_t, L, Kt, pitch, d0, B, ms, mew_s, Ks, Rfs,
Rft,Th_i,Th_o,Tc_i,Tc_o, Cp_h, Cp_c, etta, ny, H, Ce, i):
    ...
    krill_herd = [[random.uniform(0, 1) for _
    in range(10)] for _ in range(100)]
    fitness = [krill_herd_optimization(C,n1,Ds,mt,rho_t,
    rho_s, n, mew_t,L,Kt,pitch,d0,B, ms, mew_s, Ks, Rfs,
    Rft, Th_i, Th_o,Tc_i,Tc_o,Cp_h,Cp_c,etta, ny, H, Ce,
    i)for krill in krill_herd]

    for _ in range(100):
        for i, krill in enumerate(krill_herd):
            N_old = krill_herd_motion(0.1, 0.5, krill
            herd[i-1], 0.5, 0.5)
            F_food = krill_herd_foraging(0.1, 0.5, 0.5,
            0.5)
            lambda_ = krill_herd_diffusion(0.1, 0.5)
            krill_herd[i] = [krill[j] + N_old[j] +
            F_food[j] + lambda_[j] for j in range(10)]

        fitness = [krill_herd_optimization(C, n1, Ds,
        mt, rho_t,
        rho_s, n, mew_t, L, Kt, pitch, d0, B, ms, mew_s,
        Ks, Rfs, Rft, Th_i, Th_o, Tc_i, Tc_o, Cp_h,
        Cp_c, etta, ny,H,Ce,i) for krill in krill_herd]

    return krill_herd[fitness.index(min(fitness))]

def linking_function(C,n1,Ds,mt,rho_t,rho_s,n,mew_t, L,
Kt, pitch, d0, B, ms, mew_s, Ks,Rfs,Rft,Th_i,Th_o,Tc_i,
Tc_o, Cp_h, Cp_c, etta, ny, H, Ce, i):
    ...
    return krill_herd_optimization_algorithm(C,n1,Ds,mt,
    rho_t, rho_s, n, mew_t, L, Kt,pitch,d0,B,ms,mew_s,
    Ks, Rfs, Rft, Th_i, Th_o, Tc_i,Tc_o,Cp_h,Cp_c,etta,
    ny, H, Ce, i)

# Example usage
C = 0.158
n1 = 2.263
...

best_krill = linking_function(C, n1, Ds, ...)
print(f"Best krill: {best_krill}")
Error {

Cell In[9], line 466
    452 def krill_herd_motion(N_max, n,N_old,i_local,
    i_target):
    453     """
    454     Calculate the krill herd motion.
    455
    (...)
    464         float: New krill position.
    465     """
--> 466     return N_max * (n * N_old+i_local+i_target)

TypeError: can't multiply sequence by non-int of type
'float'
```

```
Code-
import random

# Define the heat exchanger model functions
def Prandtl_number(mew, Cp, Kt):
    ...

# Define the optimization algorithm
def kho_parameters():
    n = 50  # Number of krill individuals
    d = 3   # Number of dimensions
    ...
    return n, d, N_max, V_f, D_max, C_t, C_n, C_d, I_a

def initialize_krill(n, d):
    return [[random.uniform(0.1, 1.5), random.uniform
    (0.015,
    0.051), random.uniform(0.05, 0.5)] for _ in range(n)]

def calculate_fitness(krill, mt, rho_t, rho_s, n,mew_t,L,
Kt, pitch, d0, ms, mew_s, Ks, Rfs, Rft, Th_i, Th_o, Tc_i,
Tc_o, Cp_h, Cp_c, etta, ny, H, Ce, i):
    fitness = []
    for individual in krill:
        Ds, do, B = individual
        hs = Shell_Side_heat_transfer_coefficient(Kt,
        pitch, d0, Ds, B, ms, mew_s, Cp_c, Ks)
        ...
        Ctot = total_cost(Ci, Cod)
        fitness.append(Ctot)
    return fitness

def calculate_motion(krill, N_max, C_t, I_a, I):
    ...

def calculate_foraging(krill, V_f, C_n, I_a, I):
    foraging = []
    ...

def calculate_diffusion(krill, D_max, C_d, I_a, I):
    ...

def update_krill(krill, motion, foraging, diffusion):
    return [[krill[i][0] + motion[i] + foraging[i] +
    diffusion[i], krill[i][1], krill[i][2]]
    for i in range(len(krill))]

def kho_linking_function(mt, rho_t, rho_s,n, mew_t, L,
Kt,pitch, d0, ms, mew_s, Ks, Rfs, Rft, Th_i, Th_o,Tc_i,
Tc_o, Cp_h,Cp_c, etta, ny, H, Ce, i):
    n, d, N_max, V_f, D_max, C_t, C_n, C_d, I_a =
    kho_parameters()
    krill = initialize_krill(n, d)
    for I in range(I_a):
        fitness = calculate_fitness(krill, mt, rho_t,
        rho_s, n, mew_t, L, Kt, pitch, d0, ms, mew_s,
        Ks, Rfs, Rft, Th_i, Th_o, Tc_i, Tc_o, Cp_h,
        Cp_c, etta, ny, H, Ce, i)
        motion = calculate_motion(krill, N_max, C_t,
        I_a, I)
        foraging = calculate_foraging(krill, V_f, C_n,
        I_a, I)
        diffusion = calculate_diffusion(krill,D_max,C_d,
        I_a, I)
        krill = update_krill(krill, motion,foraging,
        diffusion)
    return krill

# Run the KHO algorithm
mt, rho_t,rho_s,n,mew_t,L, Kt, pitch, d0, ms, mew_s, Ks,
Rfs, Rft, Th_i, Th_o, Tc_i, Tc_o, Cp_h, Cp_c,etta,ny, H,
Ce, i = 18.80, 995, 850, 4, 0.00358, 4.88,0.13,4, 0.051,
5.52, 0.0004, 0.13, 0.00061, 0.00061,199,93.3,37.80,76.7,
2.47, 2.05, 0.9, 10, 7000, 0.12, 0.1
krill = kho_linking_function(mt, rho_t,rho_s,n,mew_t, L,
Kt, pitch, d0, ms, mew_s, Ks,Rfs,Rft,Th_i,Th_o, Tc_i,
Tc_o, Cp_h, Cp_c, etta,ny, H, Ce, i)
print(krill)

Results-
[1.28238599971764, 0.015768897423495, 0.42015856034693]
```

Table 7: Accuracy/Correctness Evaluation

| Code Generated by Paper - A | Code Generated by paper - C | Code Generated by paper - B |
|---|---|---|
| ```python
import math
import random

# Define the heat exchanger model
functions
def Prandtl_number(mew, Cp, Kt):
    Pr = (mew * Cp) / Kt
    return Pr

def total_cost(Ci, CoD):
    return Ci + CoD

# Define the optimization algorithm
def genetic_algorithm(Ds,do,B,mt, rho_t,
rho_s, n, mew_t, L, Kt, pitch, d0, ms,
mew_s, Ks, Rfs, Rft, Th_i, Th_o, Tc_i,
Tc_o, Cp_h, Cp_c, etta, ny, H, Ce, i):
    # Initialize the population
    population = []
    for _ in range(50):
        individual = {
        'Ds':random.uniform(0.1,1.5),
        'do':random.uniform(0.015,
        0.051),
        'B':random.uniform(0.05, 0.5),
        'fitness':float('inf')
        }
        population.append(individual)

    # Define the optimization parameters
    mutation_rate = 0.1
    crossover_rate = 0.5
    selection_rate = 0.5
    generations = 100

    # Run the genetic algorithm
    for _ in range(generations):
        # Evaluate the fitness of each
        individual
        ....

        # Select the fittest individuals
        population = sorted(population,
        key....

        # Replace the least fit
        individuals with the new
        offspring population = population
        + offspring

    # Return the fittest individual
    return population[0]

# Run the genetic algorithm
Ds, do, B = 1.5, 0.051, 0.5
mt, rho_t, ...= 18.80, 995, ...
best_individual=genetic_algorithm(Ds,do,
B, ...)
print('Best individual:',best_individual)
``` | ```python
import math
import random

# Define the heat exchanger model functions
...
# Define the optimization algorithm
...
def linking_function(constant, exponent,
shell_diameter, mass_flow_rate_tube,...):
        # ... (rest of the linking function code
        remains the same)

def differential_evolution(strategy, NP, F,
CR, MAXGEN, seed):
    # Initialize population
    population = []
    ...

    # Evaluate initial population
    fitness = []
    ...

    # Evolve population
    for gen in
    ...

            # Select best individual
            if trial_fitness < fitness[i]:
                population[i] = trial
                fitness[i] = trial_fitness

    return population, fitness

# Run differential evolution
strategy = "DE/best/1/exp"
NP = 70
F = 0.5
CR = 0.9
MAXGEN = 30
seed = 10
population, fitness = differential_evolution
(strategy, NP, F, CR, MAXGEN, seed)

# Print best individual
best_individual = population[fitness.index
(min(fitness))]
x1 = best_individual[0] * 2.5 + 0.25  # tube
outer diameter
x2 = best_individual[1] * 2 + 1  # tube pitch
...
print("Best individual:", x1, x2, x3, x4,
x5, x6, x7)
print("Best fitness:", min(fitness))
``` | ```python
import math
import random

# Define the heat exchanger model functions
def Prandtl_number(mew, Cp, Kt):
    Pr = (mew * Cp) / Kt
    return Pr

# Define the objective function
def objective_function(Ds, do, B):
    mt, rho_t, rho_s, n, mew_t, L, Kt, pitch,
    d0, ms, mew_s, Ks, Rfs, Rft, Th_i, Th_o,
    Tc_i, Tc_o, Cp_h, Cp_c, etta, ny, H, Ce,
    i = 18.80, 995, 850,...
    Ctot = total_cost(Ci, Cod)

    return Ctot

# Define the optimization algorithm(PSO)
def optimization_algorithm(Ds, do, B):
    # Initialize the particles
    particles = []
    for _ in range(50):
        particle = {
            'Ds':random.uniform(0.1, 1.5),
            'do':random.uniform(0.015,
            0.051),
            'B':random.uniform(0.05, 0.5),
            'velocity': [random.uniform
            (-1, 1), random.uniform(-1, 1)
            , random.uniform(-1, 1)],
            'best_position': [random.uniform
            (0.1, 1.5), random.uniform
            (0.015, 0.051), random.uniform
            (0.05, 0.5)],
            'best_cost': float('inf')
        }
        particles.append(particle)

    # Define the PSO parameters
    w = 0.9
    c1 = 2
    c2 = 2
    Vmax = 1

    # Run the PSO algorithm
    for _ in range(100):
        ....
# Run the optimization algorithm
Ds, do, B = 1.5, 0.051, 0.5
best_position, best_cost = optimization_
algorithm(Ds, do, B)
print('Best position:', best_position)
print('Best cost:', best_cost)
``` |

Table 8: Functionality Evaluation

| Code Generated by Paper - B | Code Generated by paper - C | Code Generated by paper - A |
|---|---|---|
| ```python import math import random  # Define the heat exchanger model functions def Prandtl_number(mew, Cp, Kt):     Pr = (mew * Cp) / Kt     return Pr ....  return Ci + CoD  # Define the QPSOZ algorithm def qPSOZ(Ds, do, B,...):     # Initialize the particles     particles = []     for _ in range(50):         particle = {             'Ds': random.uniform(0.1, 1.5),             'do':random.uniform(0.015,             0.051),             'B': random.uniform(0.05, 0.5),             'velocity': _cost': float('inf')         }         particles.append(particle)      # Define the QPSOZ parameters     w = 0.9     c1 = 2     c2 = 2     Vmax = 1      # Run the QPSOZ algorithm     for _ in range(100):         for particle in particles:             # Calculate the cost of the             current position             hs = Shell_Side_heat_transfer             _coefficient(Kt, pitch, d0,             particle['Ds'], particle['B'],             ms, mew_s, Cp_h, Ks) ...... # Run the QPSOZ algorithm Ds, do, B = 1.5, 0.051, 0.5 mt, rho_t, ..., = 18.80, 995, 850, ... best_position, best_cost = qPSOZ(Ds, do, B, mt,...) print('Best position:', best_position) print('Best cost:', best_cost) ``` | ```python import math import random  # Define the heat exchanger model functions def Prandtl_number(mew, Cp, Kt):     Pr = (mew * Cp) / Kt     return Pr ....     return Ci + CoD  # Define the ICA algorithm functions defcreate_initial_countries(Nvar, NCountry):     ....  def calculate_cost(country, Ds, do, B, ....i):     hs = Shell_Side_heat_transfer_coefficient     (Kt, pitch, d0, Ds, B, ms, mew_s, Cp_h)     ....     Ctot = total_cost(Ci, Cod)     return Ctot  def assimilation_policy(country, imperialist, Ds, do, B, mt, ....     return new_country  def revolution(country, Ds, do, B, mt, rho_t, rho_s, n, mew_t, L, Kt, pitch ....     return new_country  def imperialistic_competition(countries, Ds ....     return new_countries  def ICA(Nvar, NCountry, Ds, do, B, mt, ....     return best_country  # Run the ICA algorithm Nvar = 3 NCountry = 50 Ds, do, B = 1.5, 0.051, 0.5 mt, rho_t, .... Ce, i = 18.80, 995, 850, .... 0.12, 0.1 best_country = ICA(Nvar, NCountry, Ds, do, B, mt, rho_t, rho_s, n, mew_t, L, Kt, pitch, d0, ms, mew_s, Ks, Rfs, Rft, Th_i, Th_o, Tc_i, Tc_o, Cp_h, Cp_c, etta, ny, H, Ce, i) print('Best country:', best_country) ``` | ```python import math import random  # Define the heat exchanger model functi -ons def Prandtl_number(mew, Cp, Kt):     Pr = (mew * Cp) / Kt     return Pr .... return Ci + CoD  # Define the optimization algorithm def optimization_algorithm(Ds, do, B, .... Ce, i):     # Initialize the population     ....     #Define the optimization parameters     Cr = 0.5     Mr = 0.1     r = 0.96     max_iterations = 100      # Run the optimization algorithm     for _ in range(max_iterations):         .... # Select the fittest individual     fittest_individual = min     (population,     key=lambda individual:     individual['fitness'])  ....     # Return the fittest individual     return fittest_individual  # Run the optimization algorithm Ds, do, B = 1.5, 0.051, 0.5 mt, rho_t, rho_...., Ce, i = 18.80, 995, 850....0.12, 0.1 fittest_individual = optimization _algorithm(Ds, do, B, ....Ce, i) print('Fittest individual:', fittest_individual) ``` |

Table 9: Completeness Evaluation

| Code Generated by Paper - A | Code Generated by paper - B | Code Generated by paper - C |
|---|---|---|
| <pre>import math
import random

# Define the heat exchanger model functions
....
    return Ci + CoD

# Define the Cuckoo Search Algorithm (CSA)
functions
def CSA(Ds, do, B, mt, rho_t, rho_s, n,
mew_t, L, Kt, pitch, d0, ms, mew_s, Ks, Rfs,
Rft, Th_i, Th_o, Tc_i, Tc_o, Cp_h, Cp_c,
etta, ny, H, Ce, i):
    # Initialize the population
    population = []
    for _ in range(50):
        individual = {
            'Ds': random.uniform(0.1, 1.5),
            'do': random.uniform(0.015,
            0.051),
            'B': random.uniform(0.05, 0.5),
            'fitness': float('inf')
        }
        population.append(individual)

    # Define the CSA parameters
    Pa = 0.25
    alpha = 1.0

    # Run the CSA algorithm
    for _ in range(100):
        for individual in population:
            # Calculate the fitness of the
            individual
            hs = Shell_Side_heat_transfer_
            coefficient(Kt, pitch, d0,
            individual['Ds'],individual['B']
            , ms, mew_s, Cp_h, Ks)
            ...

            # Update the fitness of the
            individual
            individual['fitness'] = Ctot

        # Perform the Levy flight
        for individual in population:
            ....

# Run the CSA algorithm
Ds, do, B = 1.5, 0.051, 0.5
mt, rho_t, rho_...., Ce, i = 18.80, 995,
850....0.12, 0.1
best_individual = CSA(Ds, do, B, ....Ce, i)
print('Best Ds:', best_Ds)
...
print('Best fitness:', best_fitness)</pre> | <pre>import math
import random

#Define heat exchanger model functions
def Prandtl_number(mew, Cp, Kt):
    ....
def total_cost(Ci, CoD):
    return Ci + CoD

# Define the optimization algorithm
def genetic_algorithm(Ds, do, B, mt, rho_t,
rho_s, n, mew_t, L, Kt, pitch, d0, ms, mew_s,
Ks, Rfs, Rft, Th_i, Th_o, Tc_i, Tc_o, Cp_h,
Cp_c, etta, ny, H, Ce, i):
    # Initialize the population
    population = []
    for _ in range(50):
        individual = {
            'Ds': random.uniform(0.1, 1.5),
            'do': random.uniform(0.015,
            0.051),
            'B': random.uniform(0.05, 0.5),
            'fitness': float('inf')
        }
        population.append(individual)

    # Define the optimization parameters
    mutation_rate = 0.1
    crossover_rate = 0.5
    selection_rate = 0.5
    generations = 100

    # Run the genetic algorithm
    for _ in range(generations):
        # Evaluate fitness of each individual
        ....
            Cod = discounted_operating_cost(Co,
            ny, i)
            Ctot = total_cost(Ci, Cod)
            individual['fitness'] = Ctot

        # Select the fittest individuals
...
    # Return the fittest individual
...
    return population[0]

# Run the genetic algorithm
Ds, do, B = 1.5, 0.051, 0.5
mt, rho_t, rho_...., Ce, i = 18.80, 995,
850....0.12, 0.1
best_individual = genetic_algorithm(Ds, do, B
, ....Ce, i)

print('Best individual:', best_individual)</pre> | <pre>import math
import random

# Define the heat exchanger model functi
-ons
def Prandtl_number(mew, Cp, Kt):
    ....
def total_cost(Ci, CoD):
    return Ci + CoD

# Define the PSO algorithm functions
def particle_swarm_optimization(Ds, do,
B, mt, ....)
    ....

    # Define the PSO parameters
    w = 0.9
    c1 = 2
    c2 = 2
    Vmax = 1

    # Run the PSO algorithm
    for _ in range(100):
        for particle in particles:
            # Calculate the cost of
            current position
# Run the PSO algorithm
Ds, do, B = 1.5, 0.051, 0.5
mt, rho_t, rho_...., Ce, i = 18.80, 995,
850... .0.12, 0.1
fittest_individual = optimization_
algorithm (Ds, do, B, ....Ce, i)
print('Fittest individual:',
fittest_individual)

#But nothing described here about the
Ant Colony Optimization (ACO) technique</pre> |

Table 10: Readability and Maintainability Evaluation

| Code Generated by Paper B | Code Generated by paper A |
|---|---|
| ```python
def tube_side_reynolds_number(vt, di, mew_t):
    """Equation 3: Tube side Reynolds number"""
    return vt * di / mew_t

def tube_side_prandtl_number(mew_t, Cp_t, kt):
    """Equation 6: Tube side Prandtl number"""
    return mew_t * Cp_t / kt

def tube_side_convective_coefficient(kt, di,vt,mew_t, Cp_t, L):
    """Equation 1: Tube side convective coefficient"""
    Ret = tube_side_reynolds_number(vt, di, mew_t)
    Prt = tube_side_prandtl_number(mew_t, Cp_t, kt)

    if Ret < 2300:
        return (kt / di) * (3.657 + (0.0677 * (Ret * Prt *
        (di / L)**1.33)**0.33) / (1 + (0.1 * Prt * (Ret *
        (di / L)))**0.3))
    elif 2300 < Ret < 10000:
        ...
    else:
        return 0.027 * (kt / di) * (Ret**0.8) * (Prt**0.33)
...
def total_cost(Ci, CoD):
    return Ci + CoD
# Define the master objective function
def master_linking_function(**kwargs):
    return calculate_heat_transfer_coefficients(
        kwargs['Dt'],
        kwargs['Ds'],
        kwargs['Lt'],
        1,  # tube_side_velocity
        ...
        303  # cold_outlet_temperature
    )
# Define the optimization functions
...
# Define the fitness function
def fitness(individual):
...
# Define the genetic algorithm parameters
population_size = 100
generations = 100
mutation_rate = 0.1
crossover_rate = 0.5
# Define the design variables and their limits
design_variables = {
    'Dt': [0.01, 0.05],  # Tube diameter
    'Ds': [0.1, 1.5],  # Shell diameter
    'Lt': [0.05, 0.5],  # Tube length
}
# Initialize the population
population = []
...
# Evaluate the fittest individual using the master linking
function
results = master_linking_function(**fittest_individual)
print("Results:")
for key, value in results.items():
    print(f"{key}: {value}")
``` | ```python
# Define the heat exchanger model functions
def Prandtl_number(mew, Cp, Kt):
    Pr = (mew * Cp) / Kt
    return Pr
...
def Tube_side_heat_transfer_coefficient(C, n1, Ds, mt, rho_t, n,
mew_t, Cp, d0, L, Kt):
    di = d0 * 0.8
    Nt = num_tubes(Ds, d0, C, n1)
    vt = tube_velocity(mt, di, rho_t, n, Nt)
    Re = reynolds_number(vt, di, mew_t)
    ft = darcy_friction_factor(Re)
    Pr = Prandtl_number(mew_t, Cp, Kt)

    if Re <= 2300:
        ht = (Kt / di) * (3.657 + (0.0677 * (Re * Pr * (di / L)
        **1.33)**0.33) / (1 + (0.1 * Pr * (Re * (di / L)))**0.3))
    elif 2300 < Re <= 10000:
        ...
    else:
        ...
    return ht
...
def total_cost(Ci, CoD):
    return Ci + CoD
# Define the Cuckoo Search Algorithm (CSA) functions
def CSA(Ds, do, B,...):
    # Initialize the population
    population = []
    for _ in range(50):
        individual = {
            'Ds': random.uniform(0.1, 1.5),
            'do': ...
            'fitness': float('inf')
        }
        population.append(individual)
    # Define the CSA parameters
    Pa = 0.25
    alpha = 1.0
    # Run the CSA algorithm
    for _ in range(100):
        for individual in population:
            # Calculate the fitness of the individual
            hs = Shell_Side_heat_transfer_coefficient(Kt,
            pitch, d0, individual['Ds'], individual['B'],
            ms, mew_s, Cp_h, Ks)
            ht = Tube_side_heat_transfer_coefficient(0.158,
            ....
            Ctot = total_cost(Ci, Cod)
            # Update the fitness of the individual
            individual['fitness'] = Ctot
            ...
# Run the CSA algorithm
Ds, do, B = 1.5, 0.051, 0.5
mt, rho_t, rho_...., Ce, i = 18.80, 995, 850....0.12, 0.1
best_individual = CSA(Ds, do, B, ....Ce, i)
...
print('Best fitness:', best_fitness)
``` |

Table 11: Robustness Evaluation

| Code Generated by Paper C | Code Generated by paper B | Code Generated by paper A |
|---|---|---|
| ```python
import math
import random

# Define the heat exchanger model functions
def Prandtl_number(mew, Cp, Kt):
    Pr = (mew * Cp) / Kt
    return Pr
...

def LMTD(Th_i, Th_o, Tc_i, Tc_o):
    deltaT1 = Th_i - Tc_o
    deltaT2 = Th_o - Tc_i
    lmtd = (deltaT1 - deltaT2) / math.log
    (deltaT1 / deltaT2)
    return lmtd
...

def total_cost(Ci, CoD):
    return Ci + CoD

# Define the PSO algorithm functions
def particle_swarm_optimization(Ds, do, B,
mt, rho_t, rho_s, n, mew_t, L, Kt, pitch,
d0, ms, mew_s, Ks, Rfs, Rft, Th_i, Th_o,
Tc_i, Tc_o, Cp_h, Cp_c, etta, ny, H,Ce,i):
    # Initialize the particles
    ...
    return best_particle['best_position'],
best_particle['best_cost']

# Run the PSO algorithm
best_position, best_cost = particle_swarm_
optimization (Ds, do, B, mt, rho_t, rho_s,
n, mew_t, L, Kt, pitch, d0, ms, mew_s, Ks,
Rfs, Rft, Th_i, Th_o, Tc_i, Tc_o, Cp_h,
Cp_c, etta, ny, H, Ce, i)
print('Best position:', best_position)
print('Best cost:', best_cost)
``` | ```python
import math
import random

# Define the heat exchanger model functions
def Prandtl_number(mew, Cp, Kt):
    Pr = (mew * Cp) / Kt
    return Pr
...

def LMTD(Th_i, Th_o, Tc_i, Tc_o):
    deltaT1 = Th_i - Tc_o
    deltaT2 = Th_o - Tc_i
    lmtd = (deltaT1 / deltaT2)
    return lmtd
...

def total_cost(Ci, CoD):
    return Ci + CoD

# Define the optimization algorithm
def optimization_algorithm(Ds, do, B, mt,
rho_t, rho_s, n, mew_t, L, Kt, pitch, d0,
ms, mew_s, Ks, Rfs, Rft, Th_i, Th_o, Tc_i,
Tc_o, Cp_h, Cp_c, etta, ny, H, Ce, i):
    ...
    # Run the PSO algorithm
    for _ in range(100):
        for particle in particles:
            # Calculate the cost of the
            current position
            hs = Shell_Side_heat_transfer
            _coefficient(Kt, pitch, d0,
            particle['Ds'], particle['B'],
            ms, mew_s, Ks)
...
best_position, best_cost = optimization_
algorithm(Ds, do, B, mt, rho_t, rho_s, n,
mew_t, L, Kt, pitch, d0, ms, mew_s, Ks,
Rfs, Rft, Th_i, Th_o, Tc_i, Tc_o, Cp_h,
Cp_c, etta, ny, H, Ce, i)
print('Best position:', best_position)
print('Best cost:', best_cost)
``` | ```python
import math
import random

# Define the heat exchanger model functi
-ons
def Prandtl_number(mew, Cp, Kt):
    Pr = (mew * Cp) / Kt
    return Pr
...

def LMTD(Th_i, Th_o, Tc_i, Tc_o):
    deltaT1 = Th_i - Tc_o
    deltaT2 = Th_o - Tc_i
    lmtd = (deltaT1 / deltaT2)
    return lmtd
...

def total_cost(Ci, CoD):
    return Ci + CoD
# Define the optimization algorithm
...
def linking_function(constant, exponent,
shell_diameter, mass_flow_rate_tube,...):
    ...
    return population, fitness

# Run differential evolution
strategy = "DE/best/1/exp"
NP = 70
F = 0.5
CR = 0.9
MAXGEN = 30
seed = 10
population, fitness = differential_
evolution(strategy, NP, F, CR, MAXGEN,
seed)

# Print best individual
best_individual = population[fitness.
index(min(fitness))]
x1 = best_individual[0] * 2.5 + 0.25
# tube outer diameter
x2 = best_individual[1] * 2 + 1
# tube pitch
...
print("Best individual:", x1, x2, x3,
x4, x5, x6, x7)
print("Best fitness:", min(fitness))
``` |