# OpenReview forum: "LLM AGENTS FOR LITERATURE TO CODE CONVERSION:CASE STUDY OF HEAT EXCHANGER DESIGN"
_ICLR.cc/2025/Workshop/AgenticAI — ICLR 2025 Workshop AgenticAI Poster_

### Official Review · Reviewer_nCDe · 2025-03-03

**Rating:** 4
**Confidence:** 4

**Review:**

The manuscript presents an agentic framework designed for heat exchanger design tasks. Experiments are conducted to validate the effectiveness of the proposed approach through comparison with the HxLLM framework on a designated evaluation dataset.

Strengths
- The work attempts to address an important application domain through an agent-based approach
- Experimental comparison with a domain-specific baseline (HxLLM) is included

However, this work has several limitations.
1. The proposed framework largely adheres to established agentic paradigms without introducing significant innovations specific to heat exchanger design challenges. The work would benefit substantially from identifying and addressing unique challenges in heat exchanger design that require specialized agent architectures, or proposing domain-specific enhancements to the standard agentic framework that improve performance on this particular task.
2. The content is not self contained. For example, where is the 115 evaluation data from, how the articles are selected.
3. While comparison against HxLLM provides some insights, the experimental evaluation lacks breadth. There is no comparisons with general-purpose models (e.g., GPT variants, Claude) to establish relative performance gains. The ablation studies are absent to validate the contribution of individual components. There is limited analysis of why the proposed approach succeeds or fails in specific scenarios
4. There is no analysis of computational efficiency or latency, which is particularly important for multi-agent systems. The discussion on the trade-offs between performance and computational resource requirements is missing.

Typos:
- Line 93, the figure reference is missing.

---

### Official Review · Reviewer_BXYF · 2025-03-04
**Novel Method with Significant Impact**

**Rating:** 6
**Confidence:** 3

**Review:**

This paper presents the HxAgent framework, a novel multi-agent system that leverages LLMs to extract mathematical models from engineering literature and automatically convert them into executable code. Focusing on the design and optimization of heat exchangers, the framework utilizes a series of specialized agents (e.g., Summary Creator, TF-IDF, Planner, Designer, Optimization, Code Refiner, and Error Correction agents) to generate, refine, and validate code with minimal human intervention. The system is evaluated against a non-agentic baseline (HxLLM) across six criteria on 115 research articles, demonstrating notable improvements in performance.

The paper exhibits high technical quality and clarity by employing a robust and detailed methodology that includes well-structured pseudo-code, comprehensive mathematical formulations, and clear process diagrams. This work shows significance in automating the extraction and translation of complex mathematical models into optimized code, which would benefit the field of research.

### Pros:

- Innovative methodology that integrates multiple specialized LLM agents for targeted tasks and introduces self-reflection and RAG-based error correction.

- This paper is well-structured and the components are well-explained and demonstrated.

- Promises significant automation in literature-to-code conversion.

### Cons

- Please check the correctness of quotation marks.

- The multi-agent design might be challenging to replicate or implement in different contexts.

- Reliance on high-quality input prompts and external datasets could affect performance.

---

### Decision · Program_Chairs · 2025-03-05

Accept (Poster)